# Pannexin-1 channel inhibition alleviates opioid withdrawal in rodents by modulating locus coeruleus to spinal cord circuitry

Opioid withdrawal is a liability of chronic opioid use and misuse, impacting people who use prescription or illicit opioids. Hyperactive autonomic output underlies many of the aversive withdrawal symptoms that make it difficult to discontinue chronic opioid use. The locus coeruleus (LC) is an important autonomic centre within the brain with a poorly defined role in opioid withdrawal. We show here that pannexin-1 (Panx1) channels expressed on microglia critically modulate LC activity during opioid withdrawal. Within the LC, we found that spinally projecting tyrosine hydroxylase (TH)-positive neurons (LC^spinal) are hyperexcitable during morphine withdrawal, elevating cerebrospinal fluid (CSF) levels of norepinephrine. Pharmacological and chemogenetic silencing of LC^spinal neurons or genetic ablation of *Panx1* in microglia blunted CSF NE release, reduced LC neuron hyperexcitability, and concomitantly decreased opioid withdrawal behaviours in mice. Using probenecid as an initial lead compound, we designed a compound (EG-2184) with greater potency in blocking Panx1. Treatment with EG-2184 significantly reduced both the physical signs and conditioned place aversion caused by opioid withdrawal in mice, as well as suppressed cue-induced reinstatement of opioid seeking in rats. Together, these findings demonstrate that microglial Panx1 channels modulate LC noradrenergic circuitry during opioid withdrawal and reinstatement. Blocking Panx1 to dampen LC hyperexcitability may therefore provide a therapeutic strategy for alleviating the physical and aversive components of opioid withdrawal.

Opioid medications are prescribed for a variety of pain conditions, but stopping or reducing the use of these medications can produce debilitating withdrawal symptoms. These symptoms impact people with chronic opioid use, and some may transition from prescription opioid medications to illicit opioid drugs, increasing the risk of addiction and overdose. Opioid withdrawal is a significant barrier preventing people from discontinuing chronic prescription or illicit opioid use and misuse[1–3]. Current interventions rely on maintenance based opioid medications such as methadone or buprenorphine, requiring careful monitoring to prevent overdose, misuse, withdrawal, and relapse. These treatments do not address the aberrant autonomic hyperactivity that underlies many of the aversive withdrawal symptoms, including hypertension, insomnia, vomiting, diarrhoea, and piloerection. While non-opioid pharmacological options often involve clonidine or lofexidine, these medications also inhibit normal autonomic function, causing adverse effects and withdrawal symptoms when treatment is terminated[1–8].

Autonomic dysregulation is a feature of opioid withdrawal, and it is a major clinical problem. Attention has focused on the locus coeruleus (LC) because it is a key autonomic centre with a high density of μ

✉ e-mail: trangt@ucalgary.ca

opioid receptors[9]. Although increased LC activity is associated with opioid withdrawal, how this key brainstem region is engaged and the core circuitry responsible for aberrant autonomic output are not well defined[9–12]. Since microglial pannexin-1 (Panx1) channel activation is critical for long-term synaptic facilitation in the spinal cord during opioid withdrawal[13], we asked whether this mechanism may also have a supraspinal site of action. In examining the LC, we uncovered a specific top-down LC to spinal circuit that is crucial for the physical and aversive sequelae of opioid withdrawal. We show that the aberrant output of spinally projecting LC neurons during opioid withdrawal critically requires microglial Panx1 activation, providing a unifying opioid withdrawal mechanism in distinct spinal and supraspinal centres. Our work also provides two potential therapeutic approaches directed at blocking Panx1: the repurposing of probenecid and design of a more potent chemical entity. Collectively, our findings indicate that targeting Panx1 to prevent aberrant noradrenergic LC signalling may potentially be effective in treating opioid withdrawal and curbing relapse in opioid use and misuse.

## Results

### Microglial Panx1 channels are critical for the aversive and physical components of opioid withdrawal

To investigate the aversive mechanisms underlying opioid withdrawal, we used the conditioned place aversion (CPA) test wherein morphine dependent mice underwent naloxone-precipitated withdrawal while confined to one side of a two-chamber place conditioning apparatus[14]. Mice were first treated for 5 days with escalating doses (10–50 mg/kg) of intraperitoneal (i.p.) morphine sulfate, followed by administration of the opioid receptor antagonist naloxone (2 mg/kg; i.p.), and immediate placement into the conditioning chamber of the CPA apparatus for 30 min (Fig. 1A). To confirm that these mice underwent withdrawal, video recordings were made during conditioning and their withdrawal behaviours

were assessed. Mice treated with morphine displayed a spectrum of physical signs that were not observed in saline treated (morphine naïve) mice, including tremors, jumping, and wetdog shakes (Supplementary Fig. 1A–C)[15–17]. When allowed free access to both chambers one day after naloxone-precipitated withdrawal, morphine dependent mice spent significantly less time in the chamber previously paired with exposure to naloxone (Fig. 1B, C). By contrast, naloxone challenge in saline treated mice did not cause CPA.

The physical signs of opioid withdrawal depend on microglial Panx1[13]. We therefore asked whether microglial Panx1 also contributes to the aversive component of opioid withdrawal. To test this, we used mice with a tamoxifen induced deletion of *Panx1* in cells expressing the chemokine C-X3-C motif receptor $CX_3CR_1$ (*Cx3cr1*-Cre[ERT2]::*Panx1*[flx/flx]). Morphine treatment was initiated 28 days after tamoxifen, allowing for repopulation of peripheral cells but not central $CX_3CR_1$-expressing cells[13,15,18]. We found that Panx1-deficient mice, which display attenuated physical signs of morphine withdrawal[13], also showed significantly less withdrawal-induced CPA in contrast to their Panx1-expressing littermates (vehicle treated *Cx3cr1*-Cre[ERT2]::*Panx1*[flx/flx] mice) (Fig. 1D, E). The reduction in CPA in Panx1-deficient mice was not caused by tamoxifen per se because tamoxifen injected mice lacking inducible Cre recombinase (*Panx1*[flx/flx]) still acquired aversion to the withdrawal paired compartment (Supplementary Fig. 1D). Furthermore, Panx1-deficient mice exhibited similar distance travelled compared to Panx1-expressing mice when placed in an open chamber (Supplementary Fig. 1E), indicating that attenuated CPA to morphine withdrawal is not due to differences in locomotor activity[13]. In addition, morphine-induced increase in locomotor activity (Supplementary Fig. 1F, G) and conditioned place preference remained intact in Panx1-deficient mice (Supplementary Fig. 1H, I), suggesting that the acute psychoactive effects of morphine and the rewarding properties of morphine are not affected by the absence of microglial Panx1.

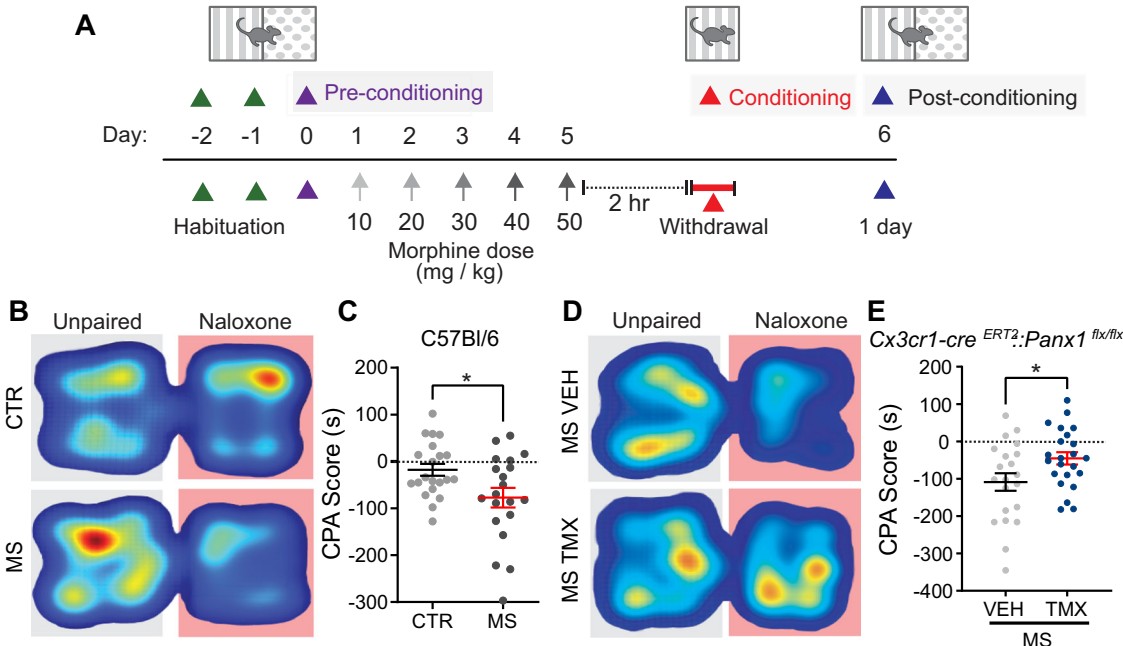

**Fig. 1 | Deletion of microglial Panx1 alleviates morphine withdrawal induced conditioned place aversion. A** Schematic of Conditioned Place Aversion (CPA) paradigm. **B** Representative heat maps of total time spent by CTR and MS treated mice during CPA during post-conditioning. **C** Quantification of naloxone-induced CPA in CTR and MS-treated mice at day 1 post-conditioning (unpaired two-sided *t*-test, *p* = 0.0196, CTR: *N* = 21 mice, MS: *N* = 20 mice). CPA score is calculated as time spent (seconds) during post-conditioning minus baseline (seconds).

**D** Representative heat maps of total time spent by MS treated vehicle (VEH) and tamoxifen (TMX) *Cx3cr1*-Cre[ERT2]::*Panx1*[flx/flx] mice during post-conditioning. **E** Quantification of Naloxone-induced CPA in *Cx3cr1*-Cre[ERT2]::*Panx1*[flx/flx] mice at day 1 (unpaired two-sided *t*-test, *p* = 0.0303) post-conditioning (MS/VEH: *N* = 21 mice, MS/TMX: *N* = 23 mice). All graphed data are presented as mean values ± SEM. **p* < 0.05. Source data are provided as a Source Data file.

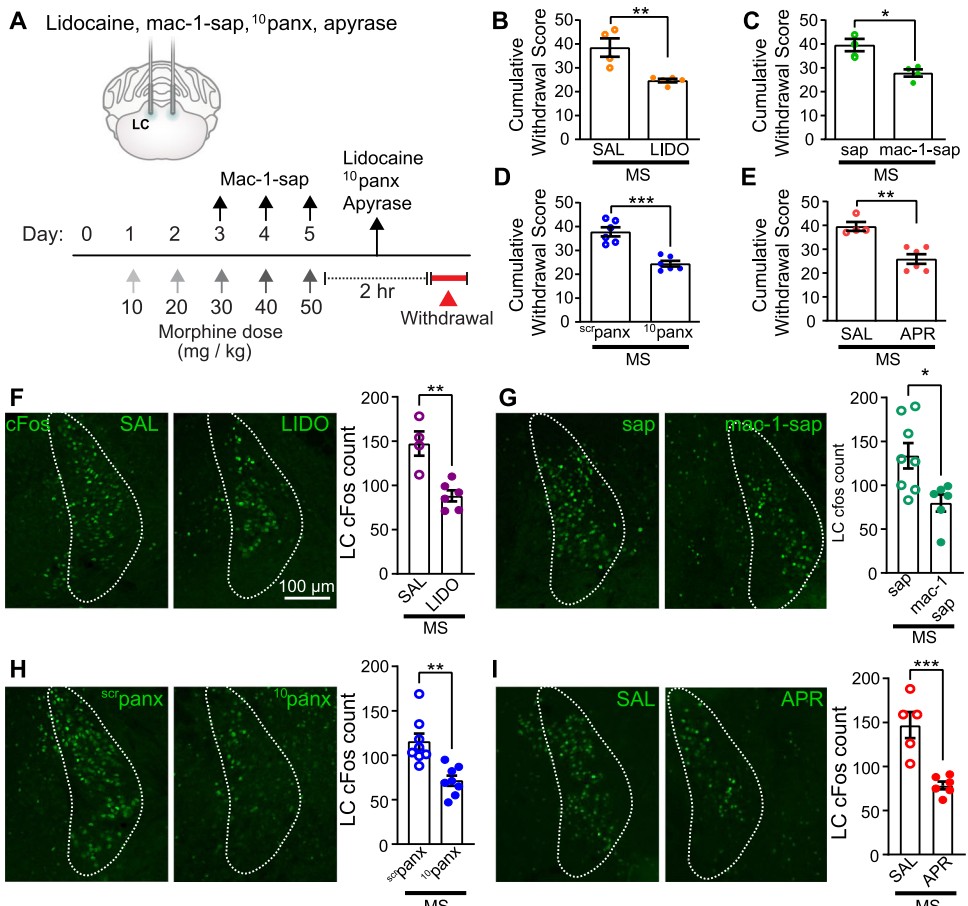

**Fig. 2 | The LC is necessary for naloxone-induced morphine withdrawal behaviours. A** Schematic depicting experimental timeline for intracerebral LC administration of lidocaine (Lido, 1%), Mac-1-saporin (Mac-1-sap, 15 μg), [10]Panx (10 μg) and apyrase (10 units). **B−E** Effects of intra-LC administration of various compounds on cumulative withdrawal scores during naloxone-precipitated withdrawal. **B** intra-LC lidocaine (Mann−Whitney two-sided test, $p = 0.0079$, SAL: $N = 4$, Lido: $N = 5$ mice), **C** mac-1-sap (unpaired two-sided $t$-test, $p = 0.010$, Sap: $N = 3$ mice, mac-1-sap: $N = 4$ mice), **D** [10]Panx (unpaired two-sided $t$-test, $p = 0.001$, [scr]Panx and [10]Panx: $N = 6$) and

**E** apyrase (APR) (Mann−Whitney two-sided test, $p = 0.0095$, Sal: $N = 4$, apyrase: $N = 6$ mice). Representative images and quantification of cFos immunoreactivity after intra-LC (**F**) Lido (unpaired two-sided $t$-test, $p = 0.0022$, SAL: $N = 4$, Lido: $N = 6$ mice), (**G**) mac-1-sap (unpaired two-sided $t$-test, $p = 0.0143$, sap: $N = 8$, mac-1-sap: $N = 6$ mice), (**H**) [10]Panx (unpaired two-sided $t$-test, $p = 0.0011$, [scr]panx and [10]Panx: $N = 8$ mice) and (**I**) apyrase (unpaired two-sided $t$-test, $p = 0.0009$, SAL: $N = 5$, apyrase: $N = 6$ mice). All graphed data are presented as mean values ± SEM. *$p < 0.05$, **$p < 0.01$, ***$p < 0.001$. Source data are provided as a Source Data file.

## LC activation during opioid withdrawal is modulated by microglia Panx1 and ATP

Since microglial Panx1 channels are critically involved in both the physical and aversive components of opioid withdrawal, we next investigated potential sites of Panx1 action within the brain. Using cFos expression as a proxy of neuronal activation, we identified several supraspinal brain regions active during opioid withdrawal. Consistent with previous studies, naloxone-precipitated withdrawal increased the number of cFos-positive (cFos+) cells in several brain regions[11,19–21], including the medial shell of the nucleus accumbens (mNAcSh), central amygdala (CeA), lateral habenula (LHb), rostral ventromedial medulla (RVM), and LC (Supplementary Fig. 2). However, in Panx1-deficient mice, the increase in cFos+ cells was suppressed in the LC and mNAcSh (Supplementary Fig. 2A, B, E), suggesting that Panx1 contributes to opioid withdrawal induced neuronal activation in these brain regions. Using RNAscope fluorescence in situ hybridisation (FISH) for *Panx1* mRNA, we detected transcripts within the LC, RVM, CeA, and mNAcSh of Panx1-expressing mice (Supplementary Fig. 3). Specific to the LC, *Panx1* mRNA was found in 48% of Iba-1 positive cells, indicating that *Panx1* is expressed on microglia within this region. Furthermore, morphine withdrawal produced a significant increase in Iba1 immunoreactivity selectively within the LC, and this

increase was abrogated in mice lacking microglial Panx1 (Supplementary Fig. 4).

Because microglial Panx1 is highly expressed in the LC and necessary for withdrawal induced cFos expression and Iba1 immunoreactivity, we examined whether interventions directed selectively at this supraspinal region could affect opioid withdrawal (Fig. 2A). We first injected lidocaine bilaterally into the LC to broadly silence neuronal activity and found that withdrawal behaviours were attenuated (Fig. 2B and Supplementary Fig. 5). We next ablated microglia in the LC with local injections of Mac-1-saporin, which produced a significant reduction (72%) in microglia after three consecutive daily injections (Supplementary Fig. 6). Depletion led to an attenuation of withdrawal behaviours, indicating that microglia in the LC play a crucial role in opioid withdrawal (Fig. 2C and Supplementary Fig. 7). By contrast, control IgG-saporin (sap) did not alter microglia expression and had no impact on withdrawal.

To block Panx1 we used [10]panx, a 10-amino acid peptide with sequence WRQAAFVDSY corresponding to residues 74-83 of the Panx1 extracellular loop 1, of which W74 and R75 are essential for gating of the Panx1 channel[22–24]. [10]panx is selective for Panx1 and does not inhibit P2X receptors[13,25]. We bilaterally injected [10]panx into the LC and found that blocking Panx1 within the LC reduced morphine withdrawal, an

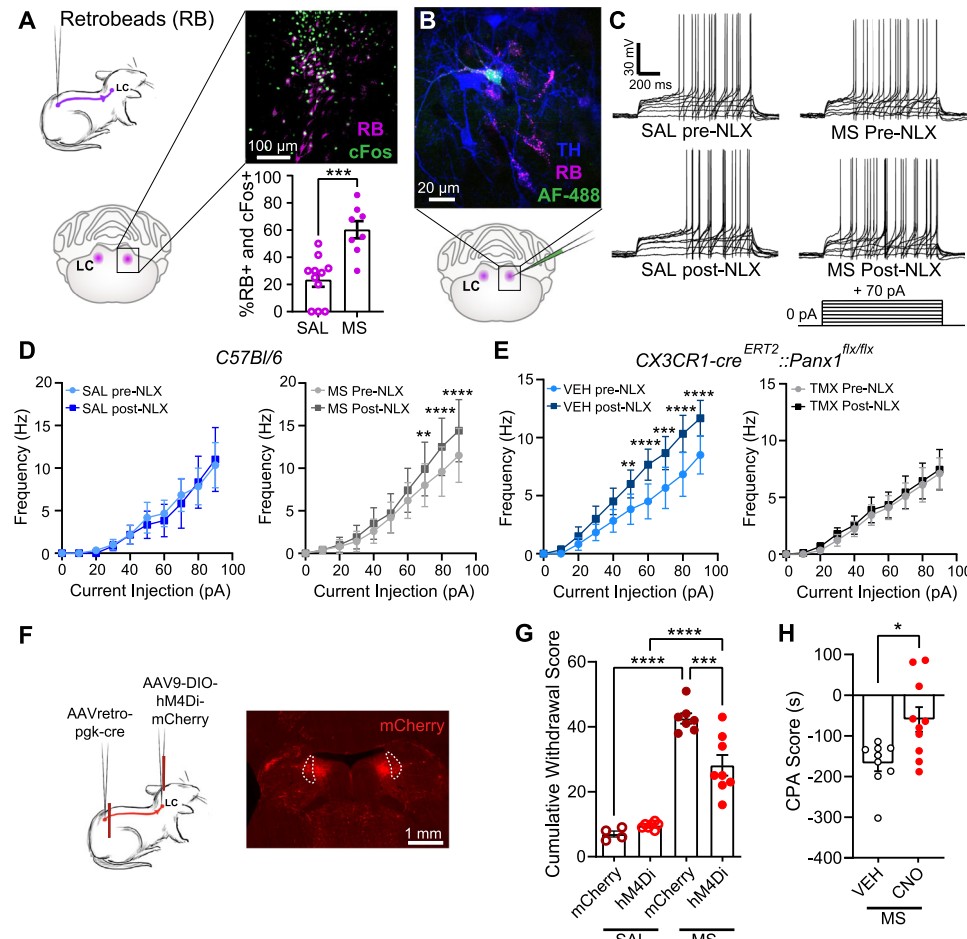

**Fig. 3 | Spinally projecting LC neurons are hyperexcitable after naloxone administration. A** Quantification of cFos and spinal retrobead (RB) colocalization after naloxone administration (unpaired two-sided $t$-test, $p = 0.0003$, SAL: $N = 11$, MS: $N = 8$ mice). **B** Representative image of electrophysiological recording in spinal RB + LC neurons. This staining was performed on 14 slices from 12 mice with similar results. **C** Example traces of LC action potential firing in response to depolarisation. **D** Left: Naloxone perfusion (10 μM) in saline (SAL) treated mice (Two-way RM ANOVA, current injection amplitude: $p < 0.0001$, pre vs. post-naloxone: $p = 0.7103$, $N = 6$ neurons/6 mice). Right: Naloxone perfusion in morphine (MS) treated mice (Two-way RM ANOVA, current injection amplitude: $p < 0.0001$, pre vs. post nalox-one: $p = 0.0381$, $N = 10$ neurons/8 mice). **E** Left: Naloxone perfusion in vehicle administered (VEH) $Cx3cr1$-Cre[ERT2]::Panx1[flx/flx] mice (two-way RM ANOVA, current injection amplitude: $p < 0.0001$, pre vs. post naloxone: $p = 0.0005$, $N = 6$ neurons/5 mice). Right: Naloxone perfusion in tamoxifen administered (TMX) $Cx3cr1$-

Cre[ERT2]::$Panx1$[flx/flx] mice (Two-way RM ANOVA, current injection amplitude: $p < 0.0001$, pre vs. post naloxone: $p = 0.3365$, $N = 9$ neurons/7 mice). **F** Schematic of chemogenetic inhibition paradigm. Example image of mCherry expression in the LC of mice with a spinal injection of AAV$_{retro}$-pgk-Cre and a bilateral injection of AAV$_9$-DIO-hM4Di-mCherry. **G** Effect of chemogenetic inhibition of spinal LC neurons with clozapine-N-oxide (CNO, 1 mg/kg, i.p.) on naloxone-precipitated withdrawal behaviours (one-way ANOVA, $p < 0.0001$, mCherry SAL: $N = 4$, hM4Di SAL: $N = 7$, mCherry MS: $N = 7$, hM4Di MS: $N = 8$ mice). **H** Effect of chemogenetic inhibition of spinal LC neurons in hM4Di mice treated with CNO (1 mg/kg, i.p.) or vehicle on CPA to withdrawal (Mann–Whitney two-sided test, $p = 0.0172$, VEH: $N = 9$, CNO: $N = 10$ mice). All graphed data are presented as mean values ± SEM. *$p < 0.05$, **$p < 0.01$, ***$p < 0.001$, ****$p < 0.0001$. Source data are provided as a Source Data file.

effect that was not observed in animals treated with the scrambled peptide [scr]panx (Fig. 2D and Supplementary Fig. 8). Given that ATP release from microglia is a consequence of Panx1 activation and ATP can modulate LC activity[26,27], we also examined the effect of locally manipulating endogenous ATP. In morphine treated mice, a bilateral LC injection of the ATP-degrading enzyme apyrase significantly reduced naloxone-precipitated withdrawal (Fig. 2E and Supplementary Fig. 9). In addition to attenuating withdrawal behaviours, each of the above LC-targeted interventions suppressed cFos expression in the LC of morphine withdrawn mice (Fig. 2F–I). Thus, both microglial Panx1 and ATP in the LC are requisite substrates for the expression of opioid withdrawal.

## Spinally projecting LC neurons are activated during opioid withdrawal

The LC sends important descending output to the lumbar spinal cord, which is a hub for opioid analgesia[28–31] and withdrawal[13,32,33]. Thus, to

investigate if naloxone-induced withdrawal is selectively engaging spinally projecting neurons (LC[spinal]), we locally injected retrobeads into the lumbar (L3-L5) spinal cord and immunostained for cFos in LC brain slices. In saline-treated mice, 23.4 ± 5.1% of retrobead labelled LC neurons were cFos+, while in morphine-treated mice, a 60.2 ± 6.3% colocalization was assessed after naloxone challenge, indicating that LC[spinal] neurons have increased activity after withdrawal (Fig. 3A). To directly assess the excitability of these neurons during withdrawal, whole cell patch clamp recordings were performed using LC brain slices acutely isolated from retrobead-injected mice treated with 5 days of morphine or saline (Fig. 3B). In LC[spinal] neurons, the average firing frequency during ramp depolarisation and minimum current injection threshold to yield action potential firing (rheobase) (Supplementary Fig. 10A) were comparable in saline and morphine treated mice. However, bath application of naloxone significantly increased action potential firing in response to depolarisation in morphine- but not saline-treated mice (Fig. 3C, D). To discern whether this

hyperexcitability is dependent on microglial Panx1 activity, LC brain slices were isolated from morphine dependent microglial Panx1-deficient (TMX MS) and Panx1-expressing (VEH MS) littermate control mice. Application of naloxone increased action potential firing in LC$^{spinal}$ neurons isolated from morphine-treated Panx1-expressing mice, but this naloxone-induced increase in activity was absent in morphine-treated Panx1-deficient mice (Fig. 3E). Notably, baseline average firing frequency during ramp depolarisation and rheobase were similar between vehicle and tamoxifen treated mice, indicating no differences due to tamoxifen treatment alone (Supplementary Fig. 10B). Therefore, increases in LC$^{spinal}$ neuron excitability during naloxone treatment are dependent on microglial Panx1.

### Chemogenetic silencing of spinally projecting LC neurons reduces aversive and physical opioid withdrawal behaviours

Our findings demonstrate that LC$^{spinal}$ neurons are hyperexcitable during opioid withdrawal and this hyperexcitability is mediated by microglial Panx1 channels. We therefore reasoned that selectively silencing activity of LC$^{spinal}$ neurons should reduce the behavioural sequelae of opioid withdrawal. To assess this, we first expressed cre selectively in spinal cord projecting neurons through injection of AAV$_{retro}$-Pgk-cre. We then bilaterally injected the Cre-dependent inhibitory designer receptor exclusively activated by designer drug (DREADD) AAV9-DIO-hM4Di-mCherry (hM4Di) into the LC (Fig. 3F). Thus, only LC$^{spinal}$ neurons would express hM4Di. Control mice received an LC injection of viral vector AAV9-DIO-mCherry, which encodes for mCherry but lacks hM4Di, and a spinal injection of AAV$_{retro}$-Pgk-Cre. We found that systemic administration of clozapine-N-oxide (CNO, 1 mg/kg) 60 min prior to naloxone challenge reduced withdrawal in morphine-treated hM4Di but not control mCherry mice (Fig. 3G) and prevented the increase in cFos+ neurons within the LC (Supplementary Fig. 11A, B). Post-hoc analysis confirmed that mCherry expression was localised within the LC (Fig. 3F, G). Using this same experimental paradigm, we found that systemic administration of CNO 60 min prior to naloxone injection was also sufficient to reduce opioid withdrawal induced CPA (Fig. 3H), without affecting locomotor activity (Supplementary Fig. 11C, D). Thus, we established that selectively silencing LC$^{spinal}$ neurons alleviates both physical and aversive opioid withdrawal behaviours.

### NE release from noradrenergic LC neurons is critical for opioid withdrawal

To define the molecular identity of LC$^{spinal}$ neurons, we immunostained LC slices fixed after electrophysiology experiments on mice with spinal retrobead injections for tyrosine hydroxylase (TH), the rate limiting enzyme responsible for synthesis of catecholamines including NE. On average, we found that each imaged LC contained $37.7 \pm 3.2$ spinally-projecting neurons (Supplementary Fig. 12A, B). Extrapolating out from this number, based on the average z-stack size (55 μm), and average depth of the LC which contained spinally-projecting neurons (approximately bregma −5.3 to −5.6), we therefore estimate that approximately 200–250 neurons within each LC of a mouse project to the spinal cord. When examining TH positivity of these retrobead+ neurons, we found that $78 \pm 4\%$ of spinal retrobead-labelled neurons within the LC are TH+, indicating that most LC$^{spinal}$ neurons are noradrenergic (Supplementary Fig. 12C). Furthermore, in wild type mice 62.9% of TH + LC neurons co-stained for cFos after naloxone-precipitated withdrawal (Fig. 4A), whereas cFos rarely co-stained with the inhibitory neuron marker glutamate decarboxylase 2 (GAD2) in Ai9 reporter mice (Supplementary Fig. 12D, E). Thus, the majority of highly active neurons (i.e. cFos+) in the LC during withdrawal are noradrenergic, but not GABAergic[34].

Since NE release is a consequence of LC activation, we measured NE levels in CSF collected from the *cisterna magna* of morphine or saline treated Panx1-expressing and microglial Panx1-deficient mice.

We found that naloxone-induced morphine withdrawal elevated NE levels in the CSF and this response was blunted in Panx1-deficient mice (Fig. 4B), suggesting that microglial Panx1 modulates NE release. To suppress NE release, we treated mice with N-(2-chloroethyl)-N-ethyl-2-bromobenzylamine (DSP4), a neurotoxin that ablates LC noradrenergic fibres without impacting peripheral sympathetic activity[35,36] (Supplementary Fig. 13A). Systemic (i.p.) injection of DSP4 depleted NE transporter (NET) expression in the spinal cord, consistent with a loss in noradrenergic terminals (Supplementary Fig. 13B–D). These DSP4-treated mice displayed attenuated withdrawal behaviours and a reduced NE response to naloxone challenge as compared with vehicle-treated morphine-dependent mice (Supplementary Fig. 13E, F). To selectively target LC$^{spinal}$ noradrenergic fibres, we then intrathecally administered DSP4. We measured the efficiency of intrathecal DSP4 in ablating LC$^{spinal}$ noradrenergic fibres by injecting AAV$_{retro}$-Pgk-cre into the spinal cord followed by LC injections of cre-dependent AAV-DIO-ChR2-eYFP to fluorescently label spinally projecting LC neurons. Next, we determined that intrathecal delivery of DSP4 reduced the number of TH+/eYFP+ (spinally projecting) neurons by 72% (Fig. 4C, D) and attenuated withdrawal behaviours (Fig. 4E), without affecting non-spinally projecting neurons within the LC. Thus, we established that selectively ablating spinally projecting TH+ neurons within the LC reduces opioid withdrawal.

### Development of probenecid and analogue compounds for treatment of opioid withdrawal and reducing risk of relapse

Having mechanistically established that microglial Panx1 channels are important for opioid withdrawal, we next asked whether targeting these channels may offer a non-opioid therapeutic approach. We therefore tested probenecid, an FDA approved broad-spectrum Panx1 inhibitor used for the treatment of gout[13]. Consistent with our previous report, systemic administration of probenecid at a higher dose of 50 mg/kg, as opposed to lower doses of 25 or 15 mg/kg (i.p.), attenuated the physical signs of naloxone-precipitated withdrawal (Fig. 5A and Supplementary Fig. 14). Furthermore, treatment with probenecid attenuated opioid withdrawal-induced CPA (Fig. 5B). Repurposing this medication for the treatment of opioid withdrawal may expedite the time to clinical translation. However, high doses of probenecid may also necessitate more frequent dosing, which could impede therapeutic compliance for people struggling with opioid use and withdrawal.

For improved potency and delivery, we synthesised the compound EG-2184 by replacing the carboxylic acid of probenecid with a tertiary hexafluoryl alcohol, which increases lipophilicity while maintaining the hydrogen bond donating properties of probenecid (Supplementary Fig. 15A). In BV-2 microglia-like cells, EG-2184 (0.01 μM) inhibited Panx1-mediated Yo-Pro-1 dye uptake at 100,000x lower concentrations than probenecid (1000 μM) (Fig. 5C, D). Similarly, EG-2184 (IC$_{50}$: 0.054 μM) was more potent than probenecid (IC$_{50}$: 74.6 μM) at inhibiting Panx1-mediated currents in human embryonic kidney-293T cells transfected with rat Panx1 (Supplementary Fig. 15B). When EG-2184 (0.1 mg/kg; i.p) was administered in mice at 100–500x lower doses than probenecid, we observed a significant reduction in the physical signs of withdrawal (Fig. 5E and Supplementary Fig. 16), decreased CPA (0.5 mg/kg) (Fig. 5F), fewer LC cFos+ cells (Supplementary Fig. 17A, B), and a blunted CSF NE release in morphine withdrawn mice (Fig. 5G). Bath application of EG-2184 (10 μM) but not DMSO vehicle control blocked the naloxone induced increase in action potential firing in LC$^{spinal}$ neurons of morphine treated mice (Fig. 5H–J). Thus, EG-2184 inhibits LC$^{spinal}$ hyperexcitability during opioid withdrawal.

In microglial Panx1-deficient mice, which display attenuated withdrawal behaviours, administration of EG-2184 (0.5 mg/kg; i.p.) did not produce a further reduction in naloxone precipitated morphine withdrawal (Supplementary Fig. 18A, B), suggesting EG-2184 mediated

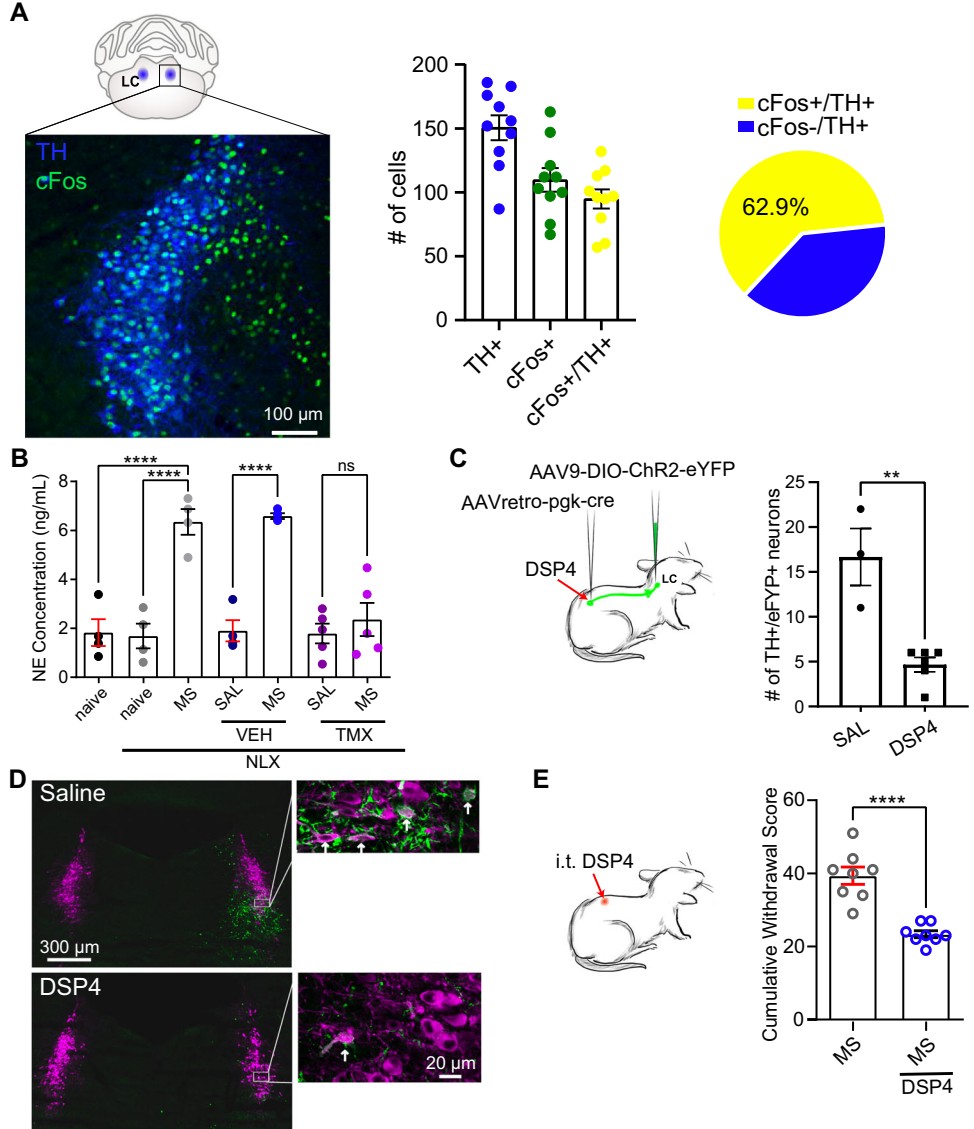

**Fig. 4 | Norepinephrine signalling in the LC drives naloxone-precipitated withdrawal behaviours. A** Quantification of tyrosine hydroxylase (TH) and cFos immunoreactivity during naloxone-precipitated withdrawal (MS NLX). $N = 10$ mice in each group. **B** Norepinephrine (NE) enzyme linked immunosorbent assay (ELISA) analysis in cerebrospinal fluid (CSF) from wild type (C57Bl/6) and Cx3cr1-Cre[ERT2]::Panx1[flx/flx] mice, after naloxone-precipitated withdrawal. C57Bl/6 mice not treated with vehicle, tamoxifen, or morphine were designated as naïve. Cx3cr1-Cre[ERT2]::Panx1[flx/flx] mice treated with vehicle or tamoxifen were designated as VEH and TMX, respectively. (One-way ANOVA, $p < 0.0001$, C57 naïve, C57-NLX, C57-MS-NLX, VEH-SAL-NLX, VEH-MS-NLX: $N = 4$, TMX-SAL-NLX and TMX-MS-NLX: $N = 5$ mice). **C** Schematic depicting experimental paradigm of bilateral injection of AAV[retro]-pgk-cre in lumbar dorsal horn and subsequent unilateral injection of AAV9-DIO-ChR2-eYFP to visualise spinally-projecting neurons. DSP4 was administered i.t. Bar graph: Number of TH+ eYFP+ neurons in the LC after i.t. saline or DSP4 administration (unpaired two-sided $t$-test, $p = 0.0066$, saline $N = 3$ slices/1 mouse, DSP4 $N = 6$ slices/2 mice). **D** Visualisation of eYFP expression in the LC in i.t. saline and DSP4 treated mice. This staining was performed on 3 slices from 1 mouse for saline treatment, and 6 slices from 2 mice for DSP4 treatment. **E** Cumulative withdrawal scores in mice treated with i.t. DSP4 (unpaired two-sided $t$-test, $p < 0.0001$, Sal-MS-NLX, DSP4-MS-NLX: $N = 8$ mice). All graphed data are presented as mean values ± SEM. *$p < 0.05$, **$p < 0.01$, ***$p < 0.001$, ****$p < 0.0001$. Source data are provided as a Source Data file.

suppression of opioid withdrawal is dependent upon its actions on microglial Panx1. Since P2X7 receptor activation is a core mechanism for opening Panx1 channels, we tested whether EG-2184 affects P2X7R-mediated Ca²⁺ responses. In BV2 microglia-like cells, BzATP (100 µM) evoked rise in intracellular [Ca²⁺] was comparable in vehicle and EG-2184 treated cells (Supplementary Fig. 18C). EG-2184 was applied at 10 nM, a concentration that blocks Panx1-mediated currents and dye flux. Thus, EG-2184 blocks Panx1 activity without off-target inhibition of P2X7R. We also established that EG-2184 (0.5 mg/kg, i.p.) did not alter responses in the open field or tail-immersion tests, indicating the compound had no effect on locomotion, anxiety, or acute pain behaviours in morphine-naïve mice (Supplementary Fig. 19A–F).

Opioid seeking and relapse are critical factors that impact chronic opioid use and misuse. We therefore tested the potential therapeutic applications of probenecid and EG-2184 in a rodent model of opioid self-administration: both extinction of opioid seeking behaviour and cue-induced reinstatement were examined (Fig. 6A). Adult male and female Long Evans rats were used for these experiments because the patency of their intravenous catheters could be better sustained. The rats were first trained to self-administer morphine in a daily 2-hour fixed ratio (FR) schedule FR1, FR2, and FR5 sessions, whereby 1, 2, and 5 lever presses, respectively, initiated morphine delivery (1.5 mg/kg/i.v. infusion). During this training, drug availability was indicated by a cue-light above the active lever. After acquisition, rats were exposed to

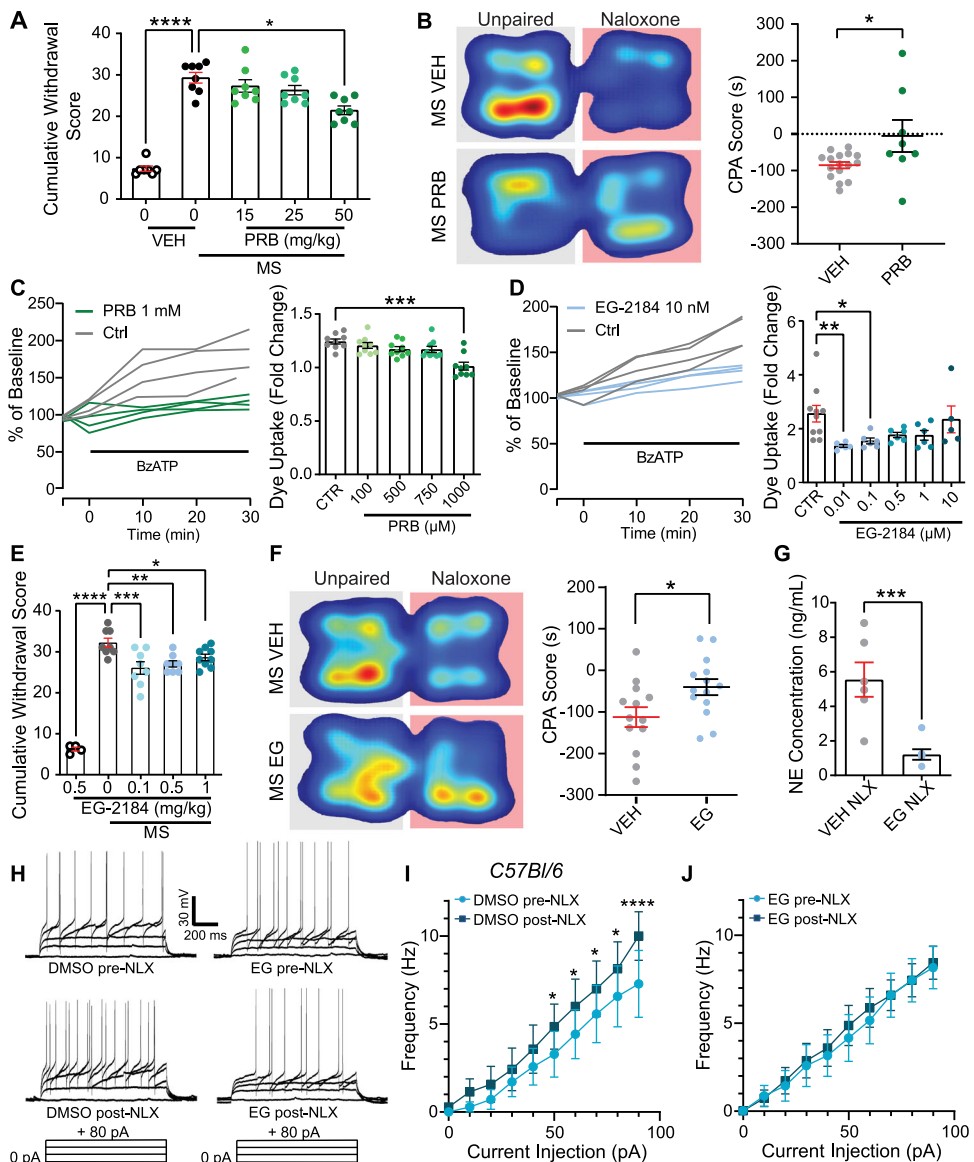

**Fig. 5 | Panx1 inhibition alleviates opioid withdrawal and blunts NE release.**
**A** Cumulative withdrawal scores in vehicle (VEH) and Probenecid (i.p., PRB) treated mice undergoing naloxone-precipitated withdrawal [Kruskal–Wallis test, $p < 0.0001$, VEH-Saline $N = 7$, VEH-morphine (MS), PRB-MS, 15 mg/kg, 25 mg/kg, 50 mg/kg: $N = 8$ mice]. **B** Conditioned place aversion (CPA) to the naloxone-paired chamber in mice treated with vehicle (VEH) or probenecid (PRB, 50 mg/kg, i.p.) (unpaired two-sided $t$-test, $p = 0.0235$, VEH $N = 16$, PRB $N = 8$ mice. **C** BzATP-evoked Yo-Pro-1 dye uptake in BV2 cells after treatment with probenecid (PRB, green, 1 mM) or control (CTR, grey) [one-way ANOVA, $p < 0.0001$, $N = 10$ cells in all groups]. **D** BzATP-evoked dye uptake after treatment with EG-2184 (blue, 10 nM) or control (CTR) [one-way ANOVA, $p = 0.0124$, CTR $N = 10$ cells, 0.01, 0.1, 0.5, 1 μM $N = 6$ cells, 10 μM $N = 5$ cells]. **E** Cumulative withdrawal scores in vehicle (VEH) and EG-2184 (i.p., EG) treated mice [one-way ANOVA, $p < 0.0001$, EG-2184: $N = 4$, MS + EG-2184 0 mg/kg, 0.1 mg/kg, 0.5 mg/kg, $N = 8$, 1 mg/kg: $N = 9$ mice]. **F** CPA to the

naloxone-paired chamber in mice treated with vehicle (VEH) or EG-2184 (i.p., 0.5 mg/kg, EG) (unpaired two-sided $t$-test, $p = 0.026$, VEH $N = 13$, EG $N = 14$ mice). **G** Norepinephrine (NE) ELISA analysis of cerebrospinal fluid CSF collected after EG-2184 treatment (Unpaired two-sided $t$-test, $p = 0.0019$, Veh-MS-naloxone, EG-MS-naloxone, $N = 6$ mice). **H–J** Naloxone perfusion (10 μM) in LC neurons pre-treated with DMSO (0.01%) or EG-2184 in DMSO (10 μM in 0.01%). **H** Example traces of LC depolarisation induced firing. **I** LC neurons pre-treated with DMSO (0.01%) [Two-way RM ANOVA, current injection amplitude: $p < 0.0001$, pre vs. post-naloxone: $p = 0.0078$, $N = 7$ neurons/7 mice]. **J** LC neurons pre-treated with EG-2184 in DMSO (10 μM in 0.01%) [Two-way RM ANOVA, current injection amplitude: $p < 0.0001$, pre vs. post-naloxone: $p = 0.521$, $N = 7$ neurons/5 mice]. All graphed data are presented as mean values ± SEM. *$p < 0.05$, **$p < 0.01$, ***$p < 0.001$, ****$p < 0.0001$. Source data are provided as a Source Data file.

10 days of extinction in which the cue light was extinguished and pressing on either lever did not initiate morphine delivery. To investigate the role for Panx1 channels in drug seeking persistency, rats received a daily i.p. injection of probenecid (100 mg/kg), EG-2184 (0.5 mg/kg), or respective vehicle (saline for probenecid, 0.1% DMSO in saline for EG-2184) 1 h before each extinction session. Across extinction sessions rats similarly decreased their number of interactions with the active lever irrespective of probenecid (Fig. 6B) or EG-2184 treatment (Fig. 6C). We then used cue-induced reinstatement, during which

the cue light associated with the morphine availability was turned back on, to assess the role of Panx1 in cue-induced drug seeking. In this context, both male and female vehicle rats significantly reinstated lever pressing (SAL + SAL, Fig. 6D and Supplementary Fig. 20). While it did not affect extinction of drug seeking (Fig. 6B), probenecid treatment, either during extinction and reinstatement or as a single injection prior to reinstatement test, prevented cue-induced drug seeking (Fig. 6D). Cue-induced reinstatement was notably mitigated when EG-2184 was administered during extinction, extinction and

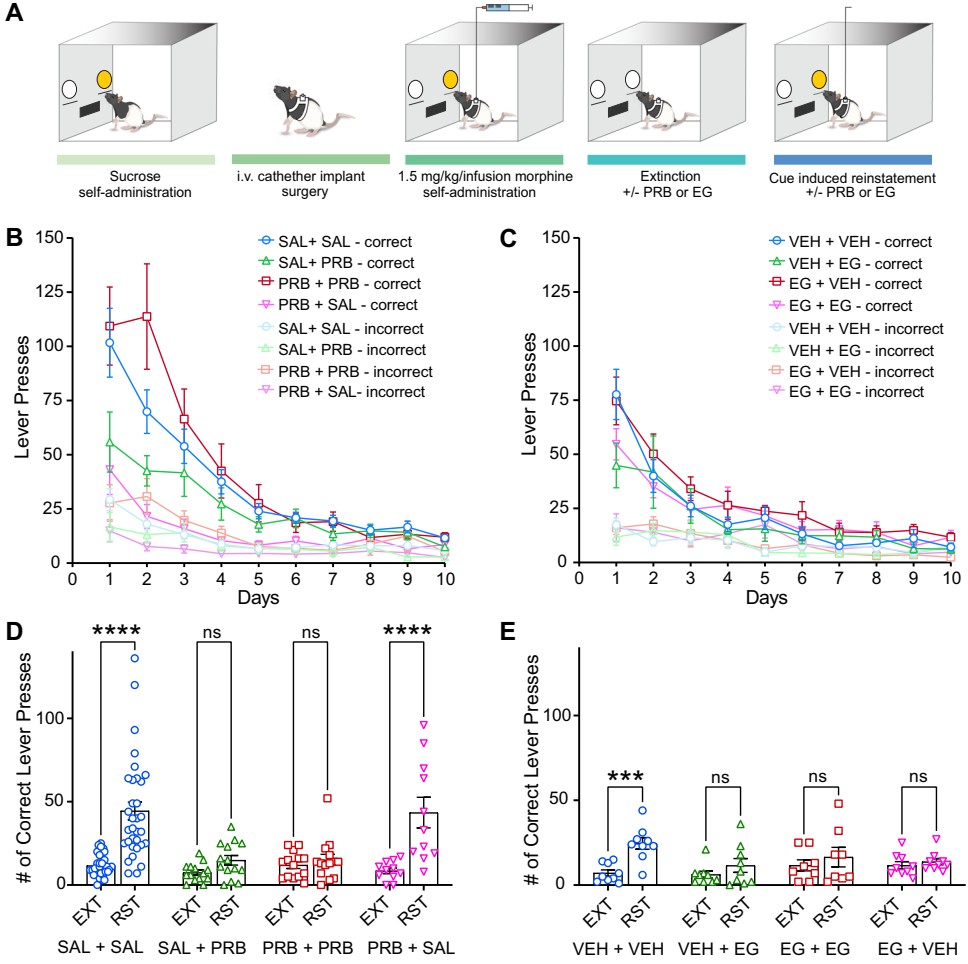

**Fig. 6 | Panx1 inhibition reduces cue-induced morphine seeking behaviours.**
**A** Schematic representation of morphine self-administration paradigm in Long Evans rats (illustration adapted from Markovic et al.[45]). **B** Progress of extinction in morphine seeking behaviours treated with probenecid (PRB, 100 mg/kg, i.p.) vs. saline (SAL) (SAL + SAL: N = 33, SAL + PRB and PRB + PRB: N = 14, PRB + SAL: N = 11 rats). **C** Progress of extinction in morphine seeking behaviours treated with EG-2184 (EG, 0.5 mg/kg, i.p.) vs. vehicle (VEH). (EG + EG: N = 8, VEH + VEH, VEH + EG, EG + VEH: N = 9 rats). **D** Effect of PRB (100 mg/kg, i.p.) administered during extinction (EXT, PRB + SAL), reinstatement (RST, SAL + PRB), or during both periods (PRB + 

PRB) on morphine cue-induced reinstatement (two-way ANOVA, time: $p < 0.0001$, PRB: $p = 0.0002$, SAL + SAL: N = 33, SAL + PRB and PRB + PRB: N = 14, PRB + SAL: N = 11 rats). **E** Effect of EG-2184 (0.5 mg/kg, i.p.) administered during extinction (EXT, EG + VEH), reinstatement (RST, VEH + EG), or during both periods (EG + EG) on morphine cue-induced reinstatement (two-way ANOVA, time: $p = 0.0005$, EG-2184: $p = 0.2776$, interaction: $p = 0.0367$, EG + EG: N = 8, VEH + VEH, VEH + EG, EG + VEH: N = 9 rats). All graphed data are presented as mean values ± SEM. ***$p < 0.001$, ****$p < 0.0001$. Source data are provided as a Source Data file.

reinstatement, or as a single injection prior to reinstatement (Fig. 6E). Thus, blocking Panx1 with probenecid or EG-2184 abrogates cue-induced reinstatement of morphine seeking.

## Discussion

In summary, we have demonstrated that microglial Panx1 modulation of spinally projecting LC noradrenergic output is a core mechanism in opioid withdrawal. Disrupting the functional connectivity between the LC and spinal cord through targeted ablation or chemogenetic silencing reduces the behavioural and neurochemical sequelae of opioid withdrawal. We therefore identify a specific top-down LC to spinal circuit, unifying two distinct centres once separately implicated in opioid withdrawal. By establishing a link between microglial Panx1 and aberrant LC[spinal] noradrenergic activity, we also provide a mechanistic explanation and potential therapeutic target for addressing the increased autonomic drive that complicates opioid withdrawal (Supplementary Fig. 21).

Our study answers an important longstanding question—how and whether the LC is dysregulated during opioid withdrawal[9–11,37,38]. In a unique population of spinally projecting neurons within the LC, we

show that intrinsic excitability is not impacted by repeated morphine treatment. However, application of naloxone unmasks a hyperexcitability in these LC[spinal] neurons, that when pharmacologically or chemogenetically silenced, reduces opioid withdrawal and CPA behaviours. The LC[spinal] recordings were conducted in preparations that were not synaptically isolated (i.e., there was no bath application of glutamatergic and GABAergic inhibitors). The hyperexcitability therefore may be caused by alterations in the intrinsic firing properties of the recorded spinally projecting LC neuron due to increased excitatory and/or decreased inhibitory synaptic input. Indeed, the LC receives inputs from and projects to many brain regions including the ventral tegmental area, nucleus accumbens, periaqueductal grey, BNST, and thalamus, which influence opioid analgesia, withdrawal, and drug seeking behaviours[9,20,39–43]. Although adrenergic output from the LC may orchestrate activity of other brain regions important for opioid withdrawal, selectively silencing LC[spinal] projections or locally inhibiting Panx1 in the LC is sufficient to attenuate the physical and aversive components of withdrawal.

Furthermore, we demonstrated that microglia within the LC are critical for withdrawal induced activation of the LC, and for physical

withdrawal behaviours. Although microglia residing within the LC were recently implicated in responses to social stress[44], evidence for LC microglial involvement in opioid withdrawal had not previously been demonstrated. We used targeted microglia using chemical depletion by Mac1-saporin and microglia specific genetic knockdown of Panx1. For Mac1-saporin experiments, mice were treated with morphine for 5 days. On days 2–4 of morphine treatment, Mac1-saporin was locally injected into the LC, resulting in microglia depletion. Thus, microglia were depleted at the time of naloxone challenge, resulting in attenuated morphine withdrawal behaviours. By contrast, in *Cx3cr1*-Cre[ERT]::Panx1[flx/flx] mice tamoxifen was administered to induce Cre recombination in CX$_3$CR$_1$ cells, causing specific conditional knockdown of Panx1. Unlike Mac1-saporin, microglia were not ablated in these experiments. We therefore waited 28 days to allow for the normal process of repopulation of peripheral CX$_3$CR$_1$ cells; central CX$_3$CR$_1$ cells (e.g. microglia) remain Panx1-deficient[18]. We previously detailed in these mice that: 1) Cre is expressed in microglia, and 2) both Panx1 expression and function are significantly reduced in microglia after tamoxifen treatment[13].

Current interventions for opioid withdrawal and opioid use disorder are limited and often rely on opioid replacement therapy. Lofexidine or clonidine are the current and most commonly prescribed non-opioid medications for managing withdrawal symptoms[6]. They act centrally on α$_2$-adrenergic receptors, which inhibit normal autonomic function resulting in adverse effects: CNS depression, respiratory depression, bradycardia, hypotension, and withdrawal symptoms can occur when treatment is terminated. Our findings provide potential therapeutic strategies for treating withdrawal through (1) the repurposing and expedited use of probenecid, a clinically approved drug with a well-established safety profile, and (2) our Panx1-inhibiting lead compound EG-2184. Although EG-2184 is more potent than probenecid at inhibiting Panx1 currents and dye flux, the effective dose of this compound and its delivery in oral or other clinical formulations will require further pharmacokinetic and pharmacodynamic optimisation. Indeed, both probenecid and EG-2184 produced a striking reduction in opioid withdrawal and suppressed cue-induced reinstatement, a model for opioid seeking and relapse, notably without producing significant adverse effects. Thus, strategies directed at blocking Panx1 to disrupt aberrant LC-NE-spinal cord signalling may be effective in treating opioid withdrawal and curbing relapse in opioid use and misuse. Finally, these mechanistic findings may also have important implications for other medications in which aberrant autonomic output of the LC is a key feature.

## Methods
### Study design
The objectives of this study were to first determine if microglial Pannexin-1 contributes to changes in locus coeruleus neuron output to the spinal cord during opioid withdrawal, and second to define whether manipulation of this circuitry, and Pannexin-1 in general can alleviate opioid withdrawal and subsequent relapse. Samples sizes for behavioural and immunohistochemical experiments were determined based on previous publications[13]. Sample sizes for CPA and electrophysiological experiments were determined based on a pilot cohort to determine relative effect size, followed by power analysis to determine likely needed N for a significant effect. Exclusions for CPA experiments were as follows: traumatic naloxone or other drug injections on the day of behaviour, fighting in cage, incorrect chamber placement in CPA experiments, absence of naloxone-precipitated withdrawal during conditioning in CPA experiments, or a preference for the naloxone-paired chamber in non-drug treated mice such that the CPA score exceeded +75. In other behavioural experiments, mice were excluded if there were traumatic naloxone or other drug injections on the day of behaviour, or if there was fighting in the cage. In all DREADD experiments, mice were excluded if there was a lack of mCherry expression in

the LC upon post-hoc analysis. In all electrophysiological experiments, cells were excluded from analysis if resting membrane potential exceeded −35 mV in neurons, or if access resistance was ≤20 or 25 MΩ (depending on neurons vs. HEK cells) at the start of the experiment or changed by more than 25% during the entire recording period. In YO-PRO uptake assays, cells were excluded if they did not respond to an ionomycin challenge at the end of the experiment. Outliers were determined in GraphPad Prism 9.4 or 10.2.0 (GraphPad Software, Inc., California), using the ROUT function, at $Q = 1\%$, and were therefore excluded from analysis within Prism.

All treatment groups were randomly assigned to animals prior to testing. Unless otherwise stated, for immunohistochemical analyses up to four brain slices were counted from each animal, and averaged such that each data point in graphs represents one animal. For electrophysiological recordings, only one to two neurons were recorded from each animal, with data points representing each neuron. Total number of neurons and animals used for each experiment are reported within each figure legend. All electrophysiological experiments were performed with the experimenter blinded either to drug condition or genotype of mouse, with unblinding occurring only after analysis. CPA experiments were performed with the experimenter blinded to expectation, but not to drug condition or genotype as pairing had to be even between groups. Furthermore, CPA experiments were analysed using an automated programme (Ethovision) to reduce any potential bias. Immunohistochemical analyses were performed with the experimenter blinded to the drug condition or genotype. Behavioural experiments (withdrawal scoring and reinstatement) were all performed with the experimenter blinded to drug condition or genotype. Yo-pro assays were all performed with the experimenter blinded to drug condition.

### Animals
All procedures were approved by the University of Calgary and Washington University Animal Care Committees, and are in accordance with the guidelines of the Canadian Council on Animal Care and the US National Institutes of Health Guide for the Care of Use of Laboratory Animals. Long Evans rats (male and female, 8–10 weeks of age) were bred in-house at Washington University[45]. All C57Bl/6 mice (male and female, 7–14 weeks of age) were obtained from the Jackson Laboratory. For some experiments, the following transgenic mouse lines were also obtained from the Jackson Laboratory: B6.129P2(Cg)-Cx3cr1[tm2.1(cre/ERT)Litt]/WganJ (JAX #021160), GAD2-cre: Gad2tm2(cre)Zjh/J) (JAX #010802), and Ai9: B6.Cg-Gt(ROSA)26Sortm9(CAG-tdTomato) Hze/J) (JAX #007909). GAD2-cre mice were crossed with Ai9 mice to generate GAD2::Ai9 mice to express mCherry in GABAergic neurons. Animals were group housed under a 12-h/12-h light/dark cycle with ad libitum access to food and water. Rat were housed at 23 °C with 27% humidity, and mice at 21 °C with 37% humidity. Animals were randomly allocated to different test groups.

**Generation of *Cx3cr1*-Cre[ERT2]::*Panx1*[flx/flx] mice.** Male and female mice with microglial specific deletion of Panx1 were generated using a Cre-loxP system. Panx1[flx/flx] homozygote mice containing flox sequences flanking exon 2 of the Panx1 gene were crossed with C57BL6/J mice expressing Cre-ERT2 fusion protein and enhanced yellow fluorescent protein (eYFP) under the *Cx3cr1* promoter (JAX #021160, B6.129P2(Cg)-*Cx3cr1*[tm2.1(cre/ERT)Litt]/WganJ). Genotype was confirmed using PCR and homozygous *Panx1*[flx/flx] and *Cx3cr1*-Cre[ERT2] mice were bred to yield *Cx3cr1*-Cre[ERT2]::*Panx1*[flx/flx] conditional knock-out mice. To induce Cre recombination, *Cx3cr1*-Cre[ERT2]::*Panx1*[flx/flx] mice were injected intraperitoneally (i.p.) with tamoxifen (1 mg/100 μL in 90% sunflower oil, 10% ethanol; Sigma) for 3 consecutive days 4 weeks before behavioural or electrophysiological experiments. Wild-type mice were littermate mice that received vehicle injections (90% sunflower oil, 10% ethanol), while tamoxifen-related effects were controlled for using

$Panx1^{flx/flx}$ littermate mice that received tamoxifen injections. As described previously by Burma et al.[13], morphine-induced antinociception is comparable, and remains intact in male and female Panx1-deficient mice.

### Behavioural testing

**Morphine dependence and naloxone-precipitated withdrawal.** Morphine sulfate (PCCA, 20-1000-25) prepared in 0.9% sterile saline solution was injected i.p. twice daily (100 μL/20 g mouse, 9 a.m. and 5 p.m., or 8 a.m. and 4 p.m.) into male C57BL/6 mice, male and female $Cx3cr1$-Cre$^{ERT2}$::$Panx1^{flx/flx}$ or $Panx1^{flx/flx}$ mice, male and female GAD::Ai9 mice (7–14 weeks, weight ranging 16–30 g, escalating doses from 10 to 40 mg/kg over four days). On day 5, mice received a morning injection of morphine (50 mg/kg) and 2 h later naloxone (2 mg/kg, naloxone hydrochloride dihydrate, Sigma N7758) to rapidly induce opioid withdrawal. Control mice received equivalent volumes of 0.9% saline and were similarly challenged with naloxone on day 5. Signs of withdrawal were recorded as previously described[46,47]. Briefly, mice were placed into a custom-built (31.5 cm × 32 cm × 61.5 cm) plexiglass chamber directly after naloxone injection. Jumping, teeth chattering, wetdog shakes, headshakes and grooming behaviours were then evaluated at 5 min intervals for a total test period of 30 min, and a standardised score of 0 to 3 was assigned to each behaviour. Salivation, piloerection and tremors or twitching were also evaluated, with 1 point given to the presence of the behaviour during each 5 min interval. All signs were counted and compiled to yield a cumulative withdrawal score. Mice were weighed before and after naloxone challenge to calculate weight loss. In all behavioural studies, experimenters were blind to the drug treatment group and genetic profile of mice.

**Conditioned place aversion.** Place conditioning was performed in a custom-built clear plexiglass apparatus (28 × 28 × 19 cm) divided into two conditioning chambers distinct in tactile and visual cues, as described previously[48]. In this experiment, one chamber was defined with stripes and plexiglass flooring, and the other with dots and metal mesh flooring. Time spent in each chamber was recorded using monochrome GigE cameras and quantified offline using EthoVision XT 11 software (Noldus). C57BL/6J, $Cx3cr1$-Cre$^{ERT2}$::$Panx1^{flx/flx}$, and $Panx1^{flx/flx}$ mice (males and females, 8–12 weeks of age) were first placed in a two chamber conditioning apparatus and allowed free access to both chambers between 15–45 min for two days to allow habituation, followed by a preconditioning day to assess baseline preference (15 min). Animals that showed a preference over 80% to one chamber were excluded from analysis. Following this, mice were treated with escalating doses of morphine[13]. Control animals received equivalent volumes of 0.9% sterile saline. Two hours after the final morphine or saline injection on day 5, mice were injected with naloxone and confined to one side of the conditioning apparatus for 30 min. Pairing to one side of the chamber was performed using a counterbalanced design by alternating the conditioned chamber[48,49]. Due to the specific conditions for grouping into the counterbalanced design, the experimenter could not be fully blinded to the experimental protocol. Instead, the experimenter was blinded to expected effect, and analysis was automated to remove potential bias. The conditioned place aversion (CPA) test then occurred 1 day post-conditioning, during which mice were allowed free access to the conditioning apparatus for 15 min. In a subset of animals, probenecid (50 mg/kg in saline, i.p., Invitrogen P36400) or EG-2184 (0.5 mg/kg in 0.1% DMSO in saline, i.p.) was injected 1 h before conditioning, with appropriate vehicle controls run within the same experimental cohort. The conditioned place aversion (CPA) score was calculated by subtracting time spent in the conditioned chamber during baseline (pre-test) from time spent in the same chamber postconditioning. In some experiments, mice were excluded from analysis for the following reasons: traumatic injection of morphine or naloxone, incorrect chamber placement, absence of

naloxone-precipitated withdrawal during conditioning, or a preference for the naloxone-paired chamber in non-drug treated mice such that the CPA score exceeded +75. For these experiments, mice were housed in reverse light-dark cycle, and experiments were performed during the dark cycle. Data for these experiments are presented aggregated by sex.

### Operant intravenous opioid self-administration

**Surgery.** All surgeries were performed under isoflurane (2.5/3 MAC) anaesthesia using sterile aseptic techniques. For intravenous (i.v.) self-administration (SA), animals were implanted with sterile catheters in the right jugular vein and connected to a harness placed over the torso. The harness allowed for minimal stress when connecting and disconnecting the animal from infusion lines. The catheter was kept patent by daily infusions of gentamicin mixture (1.33 mg/ml). To minimise post-surgical pain, animals received a daily subcutaneous (s.c.) injection of 8 mg/kg enrofloxacin and 5 mg/kg carprofen for 2 consecutive days together with carprofen chewable tablets. Behavioural experiments started 1 week after the catheter implantation.

**Opioid self-administration.** Rat self-administration was conducted using operant-conditioning chambers (Med Associates) equipped with two retractable levers with a food magazine connected to a food pellet dispenser between them as previously described[50]. Two cue lights were positioned above the levers, and one house light was positioned on the top left-hand wall. During self-administration sessions both levers (active and inactive) were extended out with white cue light turned on only above the active lever. Presses on the correct lever resulted in reward delivery and a 20 s time-out period during which the correct and incorrect levers were retracted, and cue light was turned off. Presses on the inactive lever did not cause any changes in the environment.

Animals were placed in operant boxes and exposed to fixed ratio (FR) 1 schedule of reinforcement (1 lever press results in the delivery of one food pellet) for 2 h daily (or until the rat obtained a maximum of 60 rewards during the session) for at least 5 sessions. After acquisition of the task, rats received a jugular catheter implantation (see procedure above) and recovered for a week to avoid post-surgical pain and distress. Animals were then placed in operant boxes and their harnesses were gently tethered to a drug infusion line connected to an infusion pump. Animals were exposed to a daily 2-h intravenous self-administration session, during which a press on the active lever resulted in an intravenous 1.5 mg/kg/infusion of morphine. Rats underwent 5 sessions of FR1 schedule of reinforcement, followed by 3 sessions of FR2 (2 lever presses result in reward delivery), and then 3 sessions of FR5 (5 lever presses result in reward delivery). After the last (final) FR5 session, animals were exposed to extinction sessions (1 h session daily for 10 days) during which the animals remained untethered, both levers were extended, and both the light cue above the correct lever and the infusion pump (sound) were turned off. Presses on both correct and incorrect lever had no consequences.

To assess cue-induced reinstatement, animals were placed in the operant boxes 24 h after the last extinction session, tethered to the infusion lines, and both the cue-light associated with drug availability and the sound of the infusion pump were turned back on. During the 1-hour reinstatement session no presses on either correct or incorrect levers resulted in drug infusion. Probenecid (100 mg/kg in saline, i.p.) or EG (0.5 mg/kg, i.p.) were injected using three different treatment paradigms: (1) drugs were given 1 h prior to each extinction session but not on the day of reinstatement, (2) 1 hr before reinstatement test, or (3) 1 h prior to each extinction session and before reinstatement. For reinstatement experiments, rats were group housed prior to jugular vein catheter implant surgery with two to three animals per cage on a 12/12 h dark/light cycle (lights on at 7:00 a.m.). Rats were acclimated to the animal facility holding rooms for at least 7 days before any

manipulation. After catheter implantation, animals were single housed to prevent damage to the harnesses.

**Open field test.** The open field test was performed using two different apparatuses for two different experiments. For the experiments testing for differences in vehicle vs. tamoxifen-treated *Cx3cr1*-Cre[ERT2]::*Panx1*[flx/flx] mice, the experimental conditions were as follows. Drug naïve tamoxifen and vehicle treated *Cx3cr1*-Cre[ERT2]::*Panx1*[flx/flx] mice (male and female, 8–12 weeks of age) were placed in a custom-built open-field apparatus ($28 \times 28 \times 19$ cm) devoid of texture and visual cues. Locomotion was recorded for 10 min using monochrome GigE cameras and quantified offline using EthoVision XT 11 software (Noldus). For experiments assessing potential effects of EG-2184 in naïve animals, C57Bl/6 mice (males and females, 8 weeks of age) were placed in a custom-built wooden box measuring $40 \times 60 \times 50$ cm with a clear front wall for observation[51]. The arena was divided into 12 equal squares. Animals were placed in the centre square and allowed to explore freely for 15 min. Locomotion was recorded using a video camera and quantified offline using EthoVision XT 11 software. In addition to total distance travelled, several other measures were observed during trials including rearing, time in centre zone (two central squares), defecation, and grooming. These experiments were all run during the light-phase for the mice. Experiments were run interspersing vehicle and EG-2184 treated mice to account for potential circadian effects. Data for these experiments are presented aggregated by sex.

**Tail immersion test.** Mice were acclimated to restraint in a dark 50 mL Falcon tube with the tail exposed. A water bath (Fisher Scientific, Isotemp) was maintained at 50 °C. Animals were relaxed with the tail straight and still before experiment. One cm of the distal end of the tail of the mouse was submerged into the water and a timer was started. The timer was stopped when a tail flick response was observed, and the time was recorded. A 10 s maximum cut-off time was applied to prevent tissue injury. The tail was then dried, and the mouse was returned to the cage. This was repeated 3 times and the average time was recorded as the latency. A minimum of 10 s was allowed between testing with the same animal. Data for these experiments are presented aggregated by sex.

**Retrobead injections**
Three weeks before the withdrawal paradigm, male C57/BL6J (7 weeks of age) and male/female *Cx3cr1*-Cre[ERT2]::*Panx1*[flx/flx] mice (6-8 weeks of age) underwent bilateral injection of retrobeads (Lumafluor, NC, US) into the lumbar spinal cord to visualise spinally projecting neurons. Animals were anaesthetised with isoflurane inhalation (5% induction, 2–3% maintenance on 100% oxygen) and placed on a stereotactic frame. A longitudinal incision was made along the back, muscle and bone tissue were removed to expose the lumbar region of the spinal cord. The spinal column was secured using toothed forceps and the dura was removed with a 25 gauge needle exposing the spinal cord. A glass pipette was backfilled with Retrobeads (Lumafluor, green or red), and 600–800 nL was injected in each animal via a Nanoject (Drummond Scientific, PA, US) over 6-8 sites on both sides of the spinal cord, with even dorsoventral distribution. Each injection was 50.6 nL, injected at a speed of 26 nL/s, with the pipette remaining in place for 5 min before moving to the next site. Total injection time was ~45–60 min per mouse. After the injection, muscle and skin were closed using Vicryl sutures (Ethicon, US). Each surgery lasted 1–1.5 h per animal. Perioperative pain was managed using meloxicam (Metacam, 3 mg/kg, s.c.) as per approved animal protocol.

**Chemogenetic manipulation of LC-spinal cord pathway**
All surgeries were performed under isoflurane anaesthesia (5% induction, 2–3% maintenance on 100% oxygen). In male C57/BL6J mice, (7 weeks of age) a laminectomy was performed for bilateral injections of AAV[retro]-Pgk-cre (titre: $7 \times 10^{12}$ vg/mL, Addgene, # 24593-AAVrg) into the lumbar spinal cord (600–800 nL) as described above. Two weeks later animals received a craniotomy, and bilateral injections of AAV9-hSyn-DIO-hM4Di-mCherry (titre: $1 \times 10^{13}$ vg/mL, 200 nL, 44362-AAV9) were performed in the LC (stereotaxic coordinate from Bregma: AP: −5.32 mm; LR: ±0.75 mm; DV: −4.00 mm), to inhibit spinally projecting LC neurons respectively. A subset of animals received bilateral injections of AAV9-hSyn-DIO-mCherry (titre: $1 \times 10^{13}$ vg/mL, 200 nL, 50459-AAV9) to control for effects mediated by administration of clozapine-N-oxide (CNO in 2% DMSO, 1 mg/kg, i.p., Sigma C0832). Meloxicam (Metacam, 3 mg/kg, s.c.) was administered to manage perioperative pain. Animals returned to their homecages after surgeries for 5 weeks to allow maximum viral expression, before commencing escalating doses of morphine and naloxone precipitated withdrawal. For chemogenetic manipulation, AAV9-hSyn-DIO-hM4Di-mCherry and AAV9-hSyn-DIO-mCherry animals received CNO 60 min before naloxone injections (i.p.), and physical signs of withdrawal were recorded for 30 min after. For cFos experiments, mice were perfused 90 min after naloxone administration, and brains were post-fixed for 24 h in 4% PFA at 4 °C for immunohistochemistry. For CPA experiments with chemogenetic inhibition of spinally projecting neurons, CPA was performed as described above in mice with bilateral expression of AAV9-hSyn-DIO-hM4Di-mCherry in the LC, with CNO administered i.p. 60 min before naloxone injection and chamber pairing.

**Intra-cerebral LC cannulation**
Under isoflurane anaesthesia (2–3%, 100% oxygen), male C57/BL6J mice (7 weeks of age) were mounted onto a stereotaxic frame, and the skull exposed for craniotomy (AP −5.4 mm, ML ±0.8 mm and DV −4.0 mm). A precut steel guided cannula (HRS Scientific, C235GS-5-2.0/spc) was inserted through two small holes and secured with dental cement (Geristore, 031457520). Three weeks after surgeries, animals were given escalating doses of morphine. Mac-1-saporin (15 μg, Advanced Targeting Systems, KIT-06) was injected through the cannula with a double 32 G microinjector for 3 consecutive days before systemic naloxone administration. Lidocaine (1%, Sigma, L7757-25G), apyrase (10 units, Sigma, A6535) and 10Panx (10 μg, New England Peptide, sequence: WRQAAFVDSY) were injected 1 h before naloxone administration. Solutions were dissolved in double distilled water, with a final injection volume of 200 nL.

**DSP4 injections**
For selective ablation of noradrenergic neurons, either i.p. or i.t. injections of freshly prepared N-(2-chloroethyl)-N-ethyl-2-bromo-benzylamine (DSP4, Sigma, C8417) were used. For i.p. injections, 50 mg/kg of DSP4 suspended in 150 μL 0.9% saline or 0.9% saline alone were administered. CSF collection for ELISA, behavioural, and immunohistochemical experiments were performed in mice 8 days after i.p. injection, with opioid treatment beginning on day 3 after DSP4 administration. For i.t. injections, 50 μg of DSP4 suspended in 10 μL of phosphate-buffered saline (PBS) or PBS alone were administered via a 27-gauge needle attached to a microsyringe, which was inserted between L4 and L5 vertebrae. CSF collection and behavioural experiments were performed in mice 18 days after i.t. injection, with opioid treatment beginning on day 14 after DSP4 administration. For experiments to determine depletion of spinally-projecting TH+ neurons with i.t. DSP4, we combined spinal AAV[retro]-Pgk-cre injections and LC injections of AAV9-DIO-ChR2-eYFP with i.t. DSP4 injections. Male C57Bl/6 mice (8 weeks of age) first received spinal injections of AAV[retro]-Pgk-cre (titre: $7 \times 10^{12}$ vg/mL, Addgene, 24593-AAVrg) as described above. Four weeks later, AAV9-DIO-ChR2-eYFP [titre: $7 \times 10^{13}$ vg/mL, Addgene, 20298-AAV9, pAAV-EF1a-double floxed-hChR2(H134R)-EYFP-WPRE-HGHpA] was injected unilaterally into the right LC (AP −5.31 mm; R −0.75 mm; DV −3.9 mm) as described above

(400–500 nL). DSP4 or PBS injections occurred 14 days after AAV9-DIO-ChR2 injection. Mice were then perfused for immunohistochemistry 14 days after DSP4 injection.

## LC slice electrophysiology

Male C57/BL6J or male and female *Cx3cr1*-Cre[ERT2]::*Panx1*[flx/flx] mice (8–12 weeks) were sacrificed 2–3 h after their last morphine injection (i.p.) and brainstem tissue slices were obtained as previously described[35]. Briefly, mice were euthanized by an overdose of inhaled isoflurane, followed by a cardiac perfusion of ice-cold protective NMDG solution was performed. NMDG solution contained (mM): 93 NMDG, 2.5 KCl, 1.2 NaH2PO4, 30 NaHCO3, 20 HEPES, 25 Glucose, 5 (+) Na L-ascorbate, 3 Na Pyruvate, 0.5 CaCl2·2H2O, 10 MgSO4·7H2O, 2 Thiourea, and 0.01 morphine where applicable. The brain was then surgically removed from the skull in a dish filled with ice-cold NMDG, bubbled with 95% O2/5%CO2. The frontal cortex was removed, and the brain was mounted cerebellum facing up on a vibratome disk with superglue, for coronal sectioning. Two hundred and sixty μm coronal slices were obtained via a Leica VT 1200S (Leica Microsystems, Germany), and placed into oxygenated NMDG solution at 32–34 °C for 11 min per slice. After this, slices were moved into oxygenated ACSF recording solution at 32–34 °C for an additional 30–45 min, prior to being allowed to return to room temperature (20–22 °C). ACSF recording solution contained (mM): 120 NaCl, 26 NaCO3, 25 Glucose, 2.5 KCl, 1.25 NaH2PO4, 2 CaCl2·2H2O, 1.3 MgSO4·7H2O, and 0.01 morphine where applicable.

Slices of brainstem with locus coeruleus were positively identified based on shape and presence of the fourth ventricle. Slices were placed under an upright microscope (Zeiss Axioskop2, Carl Zeiss, Germany), and secured in place by a harp. Spinally-projecting neurons within the locus coeruleus were positively identified by epifluorescence of retrobeads (Excited by ebx Xenon arc lamp (Carl Zeiss, Germany) and visualised through a tdtomato filter set) under a 40x immersion objective (W N Achroplan, Carl Zeiss, Germany). ACSF recording solution was continuously bubbled with 95% O2/5% CO2, heated to 30–32 °C by an inline solution heater (Warner Instruments, USA), and the perfusion rate was controlled to 2 mL/minute. Electrophysiological data was collected via an Axopatch 200B amplifier and digitised through a Digidata 1440 A digitiser (Molecular Devices, USA) for recording and analysis with pClamp 10.3 software (Molecular Devices, USA). Data were acquired at a sampling rate of 10 kHz and filtered at 2 kHz, with 5x output gain.

Current-clamp recordings were performed with glass pipettes (Borosilicate glass BF150-86-7.5, Sutter Instruments, USA) of tip resistance 3.5–5.5 MΩ, pulled by a DMZ Universal Electrode Puller (Zeitz Instruments, Germany). Intracellular solution contained (mM): 120 K-Gluconate, 10 KCl, 0.2 EGTA, 10 Phosphocreatine, 0.3 Na-ATP, 4 Mg-GTP, 10 HEPES, and 0.07 Alexa Fluor 488 Hydrazide (A10436, Thermo Fisher). Resting membrane potential was noted and if it exceeded −35 mV, neurons were not included for analysis. Neurons were also only included for recording if access resistance was ≤20 MΩ at the start of the experiment and did not change by more than 25% during the entire recording period. Membrane potential was maintained at −60 to −65 mV via negative current injection throughout the recording period. Junction potential was not corrected for during these experiments but is calculated to be approximately 14 mV, such that a recorded potential of −60 mV reflects an actual potential of −74 mV (Junction potential calculator, Molecular Devices, USA).

Two current-clamp protocols were performed for each neuron, in addition to a 5 min gap-free recording to monitor neuronal stability prior to naloxone perfusion. The first protocol was a ramp current protocol consisting of: 15 s sweeps with 3000 ms ramp current up to 100 pA, repeated five times to calculate an average. The second protocol was a square current protocol consisting of: 10 s sweeps with

1000 ms square current depolarisation increasing by 10 pA for 12 sweeps. Minimum current injection to yield action potential firing was defined as the rheobase, or threshold. Baseline protocols typically took 10-15 min to obtain, then wash-in of naloxone (10 μM) was performed for 15 min. Protocols were then repeated with naloxone in the recording chamber, for a total recording time of approximately 45–60 min. In experiments testing EG-2184, slices were perfused with either vehicle (0.1% DMSO in ACSF) or EG-2184 (10 μM in 0.1% DMSO ACSF) prior to whole cell breakthrough, and perfusion continued through baseline and naloxone perfusion. After recording, the slice orientation was marked with a cut in the tissue and slices were transferred to 4% PFA (EMS, 15714) for one hour fixation. Slices were then transferred to 1x PBS for long-term storage prior to post-hoc immunohistochemistry. Unless otherwise stated, all chemicals for electrophysiology were obtained from Sigma. Data for these experiments are presented aggregated by sex.

## Development of probenecid analogues

**4-bromo-*N,N*-dipropylbenzenesulfonamide (A).** To a solution of the 4-bromobenzenesulfonyl chloride (5.12 g, 20 mmol) in DCM (100 mL) was added dipropyl (6.06 g, 8.53 mL, 60 mmol) dropwise over 5 min at 0 °C under stirring. The mixture was stirred at ambient temperature for 16 h and concentrated *in vacuo*. The resulting residue was redissolved in chloroform (40 mL) and the solution was consecutively washed with water (20 mL) and brine (20 mL), and the organic extract was dried over Na$_2$SO$_4$, filtered, and concentrated *in vacuo*. The crude residue was purified by silica gel chromatography (25–40% EtOAc in Hexanes) to afford 6.30 g (98%) of the title compound: [1]H NMR (400 MHz, CDCl3) δ 7.70-7.65 (m, 4H), 3.12–3.04 (m, 4H), 1.61–1.50 (m, 4H), 0.88 (t, *J* = 7.6 Hz, 6H). All other characterisation data are in agreement with literature sources[52].

***N,N*-dipropyl-4-(2,2,2-trifluoroacetyl)benzenesulfonamide (B).** To a solution of sulfonamide **A** (3.21 g, 10 mmol) in THF (60 mL) we added a 2.5 M solution (4.8 mL, 12 mmol) of *n*-BuLi dropwise over 5 min at −78 °C. After 30 min, methyl trifluoroacetate (1.92 g, 15 mmol) was added over 5 min at −78 °C. The reaction mixture was allowed to stir and warm to ambient temperature over 30 min and was then quenched by addition of a saturated NH$_4$Cl solution (aq, 20 mL), water (20 mL), and diethyl ether (30 mL) with vigorous stirring for 15 min. The organic extract was isolated and the aqueous solution was extracted with diethyl ether (40 mL) followed by chloroform (40 mL). The combined organic extracts were washed with brine, dried over Na$_2$SO$_4$, filtered, and concentrated *in vacuo*. The crude residue was purified by silica gel chromatography (25% EtOAc in Hexanes) to afford 3.69 g (98%) of a mixture of the intermediate trifluoro methyl ketone and its hydrate in 1:2 ratio respectively. The mixture was redissolved in anhydrous chloroform (80 mL) and stirred with activated 4 Å molecular sieves for 48 h to fully convert all the material to the trifluoro methyl ketone (monitored by [1]H NMR). The solution was filtered and concentrated *in vauo* to furnish 2.83 g (84%) of the title compound and was used without additional purification: [1]H NMR (400 MHz, CDCl3) δ 8.20 (d, *J* = 8.8 Hz, 2H), 8.00 (d, *J* = 8.8 Hz, 2H), 3.16–3.10 (m, 4H), 1.65–1.51 (m, 4H), 0.90 (t, *J* = 7.6 Hz, 6H). Hydrated trifluoromethyl ketone: [1]H NMR (400 MHz, CDCl3) δ 7.79 (d, *J* = 8.4 Hz, 2H), 7.58 (d, *J* = 8.8 Hz, 2H), 4.32 (s, 1H), 3.03–3.00 (m, 4H), 1.62–1.48 (m, 4H), 0.89 (t, *J* = 7.6 Hz, 6H).

**4-(1,1,1,3,3,3-hexafluoro-2-hydroxypropan-2-yl)-N,N-dipropylbenzenesulfonamide (EG-2184).** A solution of the trifluoromethyl ketone **B** (2.1 g, 6.2 mmol) in dimethoxyethane (35 mL) was cooled to 0 °C, and trifluoromethyltrimethylsilane (1.27 g, 9 mmol) was added followed by CsF (0.16 g, 1 mmol). The mixture was stirred at 0 °C for 1 h followed by stirring at ambient temperature for an additional

hour. The reaction mixture was concentrated *in vacuo*, the resultant residue was redissolved in THF (40 mL) and cooled to 0 °C and treated with solid tetrabutylammonium fluoride (2.08 g, 7.44 mmol). The reaction mixture was allowed to stir and come to ambient temperature over 30 min. The reaction was quenched by addition of a 1.2 M HCl solution to until a pH of 6-7 was obtained. The reaction mixture was concentrated *in vacuo* and the resultant residue was stirred with water (20 mL) and diethyl ether (40 mL) for 5 min. The organic extract was isolated, and the aqueous mixture was extracted consecutively with ethyl acetate (30 mL) and chloroform (30 mL). The combined organic extracts were washed with brine, dried over $Na_2SO_4$, filtered, and concentrated *in vacuo*. The crude residue was purified by silica gel chromatography (15-25% EtOAc in Hexanes) to afford 1.96 g (77%) of the title compound. Recrystallisation from hexanes/chloroform gave pure product as fine needles with mp: 99-100 °C (hexanes/CHCl$_3$); $^1$H NMR (400 MHz, CDCl$_3$) δ 7.89 (s, 4H), 3.94 (s, 1H, OH), 3.16 – 3.10 (m, 4H), 1.63 – 1.53 (m, 4H), 0.89 (t, $J$ = 7.6 Hz, 6H); $^{13}$C NMR (100 MHz, CDCl3) δ 142.2 (C), 133.5 (C), 127.5 (CH), 127.1 (CH), 122.4 (d, $J$ = 288 Hz, CF$_3$), 50.1 (CH$_2$), 22.0 (CH$_2$), 11.1 (CH$_3$); $^{19}$F NMR (376 MHz, CDCl$_3$) δ −75.4; IR (film, cm$^{-1}$) 3386, 2967, 2937, 2877, 1270, 1214, 1150, 1084, 978, 934, 732, 706, 599; HRMS (ESI) $m/z$ calcd. for [C$_{15}$H$_{19}$F$_6$NO$_3$S + Na]$^+$ = 430.0882, found 430.0892.[45]

### HEK-293T patch-clamp electrophysiology

**Cell culture and transfection.** Human embryonic kidney-293T cells (293T) were purchased from ATCC and routinely maintained in the laboratory. Adherent 293T cells were grown in Dulbecco's modified Eagle's medium (Thermo Fisher) supplemented with 10% fetal bovine serum (Thermo Fisher, A5256701) and 1% penicillin-streptomycin (10,000 U/mL; Thermo Fisher) at 37 °C in a 5% CO$_2$ humidified growth incubator. Cells between the passages 6-20 were used for experiments. For patch-clamp recordings, 293T cells reaching 40−60% confluency on 35 mm culture plates were transiently transfected using Lipofectamine 2000 (Thermo Fisher) 36−48 h before experiments, using the manufacturer's protocol. *Rattus norvegicus* Pannexin-1 (rPanx1) complementary DNA (cDNA) was cloned between BamHI and SalI sites in a pRK5 expression vector (pRK5-rPanx1). Enhanced green fluorescent protein (EGFP) plasmid (pEGFP-C1) was originally commercially purchased (Addgene, discontinued). We transfected 293T cells with 2.5 μg pRK5-rPanx1 and 0.5 μg pEGFP-C1 cDNAs at a mixed ratio of 5:1 to identify positively transfected cells.

**Electrophysiology.** On the day of experiments, transiently transfected 293T cells overexpressing rPanx1 and EGFP cDNAs were seeded onto poly-d-lysine coated glass coverslips at least 2 h before recordings. Cells were maintained at 30 °C throughout the duration of each recording and voltage-clamped in whole-cell configuration using a Multiclamp 700B amplifier (Axon Instruments). Data were acquired using Clampex (v.10) software and an Axon Digidata 1550A digitiser (Axon Instruments) at 10 kHz, with currents analysed offline in Clampfit software (v.10.7). Patch pipettes were pulled from 1.5/0.86 mm [outer diameter/inner diameter] borosilicate glass (Sutter Instrument) using a P-1000 Micropipette Puller (Sutter Instrument) with a resistance of 3−5 MΩ. Patch pipettes contained (in mM) 4 NaCl, 1 MgCl$_2$, 0.5 CaCl$_2$, 30 TEA-Cl, 100 CsMeSO$_4$ 10 EGTA, 10 HEPES, 3 ATP-Mg$^{2+}$, and 0.3 GTP-Tris, pH = 7.3 (with KOH), ~290 mosmol/liter (solutes purchased from Sigma). Recordings were performed on a Zeiss Axio Observer Z1 inverted epifluorescence microscope using a 40x/0.6 air-immersion objective (Zeiss), 470 nm light-emitting diode (LED) (Zeiss) and a 38 HE filter set (Zeiss) to visualise EGFP signal. Cells were under continuous bath perfusion with extracellular solution containing (in mM) 140 NaCl, 3 KCl, 2 MgCl$_2$, 2 CaCl$_2$, 10 D-(+)-Glucose, and 10 HEPES, pH = 7.3 (with NaOH), ~305 mosmol/liter (solutes purchased from Sigma). Drug concentrations for patch-clamp experiments were selected from results obtained in the dye-uptake assay. Compounds were dissolved in DMSO (Sigma) to make a 10 mM stock solution and diluted to final concentrations in extracellular solution. Final concentrations of DMSO in extracellular solutions ranged from 0.001 to 0.1% v/v. Cells were voltage clamped at −60 mV in whole-cell configuration and exposed to a 300 ms voltage ramp (−80 to +80 mV) before stepping back to −60 mV to record rPanx1 voltage-sensitive currents. Baseline currents were measured in extracellular solution before perfusion of drug compounds in incrementally increasing concentrations. If access resistance eclipsed ≥ 25 MΩ, cells were discarded from analysis. Note that the full inhibition of transfected current by maximum doses of Probenecid represents 100% block of Panx1 (expressed as % inhibition). The remaining current is comprised of other ion channels in the 293T cells that are insensitive to the Panx1 blockers. Unless otherwise indicated, all chemicals were obtained from Sigma.

### Immunohistochemistry (cFos, Iba1, TH)

Male C57/BL6J, male and female *Cx3cr1*-Cre$^{ERT2}$::*Panx1*$^{flx/flx}$, and male and female GAD::Ai9-mCherry mice were overdosed with isoflurane within 2 h of naloxone or saline injection. Data for these experiments are presented aggregated by sex. Animals were transcardially perfused with phosphate buffered saline (PBS) followed by 4% paraformaldehyde (PFA, EMS, 15712). Brains were postfixed in 4% PFA and cryoprotected in 30% sucrose with sodium azide (Sigma, S2002). The brainstem was suspended in clear frozen section compound (VWR) on dry ice and cryosectioned into 40 μm free-floating coronal sections, washed and blocked in donkey serum (3% in 0.3% triton PBS, Sigma D9663) for 1 h. After, they were incubated overnight in primary antibody, followed by secondary antibody for 2 h (Table 1). Slices were then washed in PBS three times, with 1:1000 DAPI added in the second wash, and mounted on Superfrost slides (VWR, 48311-703). Slides were then mounted with Fluoromount solution (Sigma, F4680) and coverslipped (VWR). Images were obtained using an Olympus VS110 System Macro Slide Scanner (20x air objective, 0.75 NA) and A1R Nikon multiphoton microscope (20x water immersion objective, 0.95 NA). Quantification of cFos, TH, Iba1 immunoreactivity and retrobead expression was performed using ImageJ (NIH). Immunohistochemistry performed on fixed tissue slices from electrophysiology followed the same procedure as above, with no cryopreservation or cryosectioning. Images for these experiments were obtained using a Leica TCS SP8 confocal microscope equipped with a white-light confocal laser and HyD detectors (5x air objective, 0.15 NA and 20x air objective, 0.75 NA), and optimised in ImageJ (NIH).

### In situ hybridisation (RNAScope)

In situ hybridisation (ISH) was performed using the RNAscope singleplex assay (Advanced Cell Diagnostics). Mice were euthanized by an

### Table 1 | Summary of antibodies

| Primary antibody | Secondary antibody |
|---|---|
| rabbit anti-cFos (ab190289, Abcam, lot#GR3379960-1), 1:500 | Donkey anti-rabbit Alexa 488, (A21206, Invitrogen, lot#2289872), 1:500 |
| sheep-anti-TH (AB1542, Millipore, lot#3091702), 1:500 | Donkey anti-sheep Alexa 647, (A21448, Invitrogen, lot#2155286), 1:500 |
| rabbit anti-Iba1 (019-19741, Wako, lot#LEF4660), 1:500 | Donkey anti-rabbit Alexa-488, (A21206, Invitrogen, lot#2289872), 1:1000<br>Donkey anti-rabbit Alexa-647, (A31573, Invitrogen, lot#2420695), 1:500 |
| Mouse anti-NET (ab211463, Abcam, lot#GR3273270-2), 1:500 | Donkey anti-mouse Alexa-568 (abcam, ab175472, lot#GR3213513-1), 1:500 |

overdose of inhaled isoflurane, followed by cardiac perfusion with ice-cold PBS and then 4% PFA. Brains were removed and post-fixed with 4% PFA overnight. Brains were next processed using a TP1020 Leica Tissue Processor (Leica Biosystems, Germany), dehydrated in 70% ethanol and embedded in paraffin. Brain slices were obtained at 10 μm and mounted directly onto Superfrost slides. Paraffin removal, pre-treatment and processing were performed using the standard RNAscope singleplex protocol[53]. Briefly, slides were baked at 60 °C for 1 h, deparaffinized, treated with $H_2O_2$ at RT for 10 min, boiled in target retrieval solution for 15 min and permeabilized with protease for 30 min. Samples were then hybridised with the *Panx1* probe (Mm-Panx1, #316321) for 2 h at 40 °C. Slides were then washed and hybridised with amplifiers and visualised with Fast Red. All probes and solutions were purchased from ACDBio (California, US). Immediately after this process, slides were blocked with 10% NDS with 0.1% triton in TBS for 30 min. at room temperature and immunohistochemistry against Iba1 was performed as described in the above immunohistochemistry section.

### ELISA
CSF was extracted from adult male C57/BL6J, and adult male and female $Cx3cr1$-Cre$^{ERT2}$::*Panx1*$^{flx/flx}$ mice. Briefly, mice were anaesthetised with isoflurane (5% induction, 2–3% maintenance on 100% oxygen) and placed on a stereotaxic frame. The head and neck were shaved, and an incision was made to expose the *cisternae magna*. A glass capillary tube was then inserted past the *dura mater* and CSF was collected directly into the capillary as described previously[54]. Approximately 5 μL was collected from each mouse.

Norepinephrine (NE) content in cerebrospinal fluid (CSF) was measured with NE ELISA kit (Abnova, KA1877). NE was extracted using a cis-diol-specific affinity gel from undiluted CSF samples (10 μL), acylated and then converted enzymatically. The antigen was bound to the solid phase of the microtiter plate. The derivatized standards, controls, samples and the solid phase bound analytes competed for a fixed number of antibody binding sites. After the system was in equilibrium, free antigen and free antigen-antibody complexes were removed by washing. The antibody bound to the solid phase was detected by an anti-rabbit IgG-peroxidase conjugate using 3,3′,5,5′-Tetramethylbenzidine (TMB) as a substrate. The reaction is monitored at 450 nm. Quantification of unknown samples is achieved by comparing their absorbance with a standard curve prepared with known standard concentrations.

### Yo-PRO-1 dye uptake
Following 5 days of morphine or saline treatment, BV2 cells were incubated in ECS containing either YO-PRO-1 dye (2.5 μM, Invitrogen, Y3603). After a 5 min baseline recording, cells were stimulated with BzATP (150 μM, Alomone, A-385) and dye-uptake was recorded for 30 min. Cell viability was assessed immediately after a 30 min recording by application of ionomycin (1 μM, Sigma, I9657). Cells that did not respond to ionomycin were excluded from the analysis. YO-PRO-1 dye fluorescent emission (491/509) was detected at 37 °C using a FilterMax F5 plate reader (Molecular Devices). Drugs used included probenecid (1 mM in ECS, Life Technologies) and EG-2184 (final concentration of 0.1% DMSO in ECS, University of Calgary). Drugs were bath applied in ECS containing YO-PRO-1 dye and incubated at 37 °C for 10 min prior to BzATP stimulation. Fluorescent emission at 30 min post-BzATP application was calculated as an average of the entire imaging field as percent change from baseline, and responses at 30 min were averaged over multiple independent experiments. Individual traces of BV2 YO-PRO-1 dye uptake were taken at 10 min intervals and represent the average response of all BV2 cells in the recorded view from a subset of experiments.

### Calcium imaging
BV2 microglia-like cells were incubated for 30 min with the fluorescent $Ca^{2+}$ indicator dye Fura-2 AM (2.5 μM, Molecular Probes, F1221) in ECS containing 140 mM NaCl, 5.4 mM KCl, 1.3 mM $CaCl_2$, 10 mM HEPES, and 33 mM glucose (pH 7.35, osmolarity 315 mOsm)[55,56]. After fluorophore loading, cells were rinsed with ECS and stimulated with BzATP (100 μM). Experiments were conducted at room temperature using an inverted microscope (Nikon Eclipse Ti C1SI Spectral Confocal) and the fluorescence of individual microglia was recorded using EasyRatioPro software (PTI). Excitation light was generated from a xenon arc lamp and passed alternatingly through 340 or 380 nm bandpass filters (Omega Optical, VT, USA). The 340/380 fluorescence ratio was calculated after baseline subtraction. Unless otherwise indicated, all chemicals were obtained from Sigma.

### Data analysis and statistics
In all experiments, comparisons were first tested for normality (Shapiro–Wilk test) to ensure correct statistical analyses were performed. Where comparisons between two different data sets were performed, unpaired t-tests were performed for Gaussian distributed data, and Mann–Whitney t-tests were performed for non-Gaussian data. For comparisons of paired data sets, paired t-tests were performed for Gaussian distributed data, and Wilcoxon matched pairs signed rank test for non-Gaussian data. For comparisons between three or more sets of data, one-way ANOVAs were performed with Holm–Sidak post hoc tests (Kruskal–Wallis for non-Gaussian data sets). Where data had two separate variables, two-way ANOVAs were used to assess the effects of each variable, with Holm–Sidak post hoc tests. For two-way ANOVAs on non-Gaussian data Dunn's post-hoc tests were performed instead. The exact test performed for each experiment is reported within each figure legend.

Opioid self-administration experiments were performed at least twice, including each treatment condition to prevent an unspecific day/condition effect. All data are presented as mean ± SEM and analysed using GraphPad Prism 9.4 or 10.2.0 (GraphPad Software, Inc., California). For all statistical tests $*p < 0.05$, $**p < 0.01$, $***p < 0.001$, $****p < 0.0001$.

### Reporting summary
Further information on research design is available in the Nature Portfolio Reporting Summary linked to this article.

## Data availability
Source data are provided with this paper.

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

## Acknowledgements

We thank Dr. Dan Littman (HHMI, NYUSM) and Dr. Wen-Biao Gan (NYUSM) for generously providing breeding pairs for the Cx3cr1-Cre[ERT2] mouse colony and Dr. Frank Visser for mouse genotyping. We also thank Ms. Barbe Zochodne for comments on the manuscript, Dr. Catherine Cahill for providing expertise with the condition place aversion experiments, and the HBI Advanced Microscopy Platform Core Facility for providing access to the Nikon C1S1 confocal and A1R multiphoton microscopes used for imaging. T.T. was supported by Canadian Institute of Health Research (CIHR) (PJT-173553; PJ8-169697), Natural Sciences and Engineering Research Council of Canada (NSERC) (RGPIN06289-2019), Vi Riddell Program for Pediatric Pain, and a Campus Alberta Neuroscience Entrepreneurship Grant. G.W.Z. was supported by CIHR (PJT-173553), and a Canada Research Chair. D.J.D. was supported by NSERC (RGPIN-2017-04117) and a Campus Alberta Neuroscience Entrepreneurship Grant. J.A.M. was supported by U.S. National Institute of Health (NIH) (R01-DA041781, R01-DA042499, R01-DA045463). N.M. was supported by NIH (R21-DA055057) and the McDonnell Center for Cellular and Molecular Neurobiology. C.H.T.K. was supported by a CIHR postdoctoral fellowship, a Louise and Alan Edwards Foundation postdoctoral fellowship, and a Hotchkiss Brain Institute (HBI) postdoctoral fellowship. E.K.H. was supported by a CIHR postdoctoral fellowship, a Spinal Cord Nerve Injury and Pain (SCNIP) postdoctoral fellowship, and an Eyes High postdoctoral fellowship. N.E.B. was supported by a CIHR doctoral scholarship, and an Alberta Innovates scholarship. N.V.D.H. was supported by a HBI postdoctoral fellowship, and an Alberta Innovates postdoctoral fellowship. J.C.P. was supported by a SCNIP postdoctoral fellowship. E.G. was supported by an Alberta Innovates scholarship. J.P. was supported by an Alberta Innovates scholarship and a NSERC scholarship. R.D. was supported by a CIHR scholarship. We thank Ms. Mimi Tam for illustrations.

## Author contributions

Conceptualisation: C.H.T.K., E.K.H., N.E.B., and T.T. Methodology: C.H.T.K., E.K.H., N.E.B., and T.T. Experiments: C.H.T.K., E.K.H., N.E.B., T.M., N.M., H.J.Y., K.K., Y.K., C.F., R.D., E.G., N.v.d.H., J.C.P., Z.Z., C.L.A., and S.S.H. Design and synthesis of EG-2184: K.N., E.G., J.P., and D.J.D. Data analysis and visualisation: C.H.T.K., N.E.B., E.K.H., T.M., N.M., K.K., Y.K., C.F., E.G., N.v.d.H., J.C.P., Z.Z., C.L.A., S.S.H., and B.B.A. Supervision and funding acquisition: T.T., R.J.T., D.J.D., J.A.M., and G.W.Z. Writing—original draft: C.H.T.K., E.K.H., and T.T. Writing—review & editing: all authors contributed to reading and editing of the manuscript.

## Competing interests

T.T., D.J.D., and R.J.T. are co-founders of AphioTx Inc. Patents have been granted for the use of probenecid in opioid withdrawal (T.T., N.E.B.). Patents are pending for EG-2183 and related compounds for use in opioid withdrawal (T.T., D.J.D., C.H.T.K., E.K.H., N.E.B., E.G., K.K., K.E.N., C.F., E.G., and J.W.P. Specific patent details are provided in the Disclosure of Potential Competing Interest. G.W.Z. is co-founder and CSO of Zymedyne Therapeutics. All other authors have declared that no conflict of interest exists.

## Additional information

**Charlie H. T. Kwok**[1,2,8], **Erika K. Harding**[1,2,3,8], **Nicole E. Burma** ®[1,2,8], **Tamara Markovic**[4], **Nicolas Massaly**[4,5], **Nynke J. van den Hoogen** ®[1,2], **Sierra Stokes-Heck** ®[1,2], **Eder Gambeta**[3], **Kristina Komarek** ®[1,2], **Hye Jean Yoon** ®[4], **Kathleen E. Navis**[6], **Brendan B. McAllister** ®[1,2], **Julia Canet-Pons**[1,2], **Churmy Fan**[1,2], **Rebecca Dalgarno**[1,2], **Evgueni Gorobets**[6], **James W. Papatzimas** ®[6], **Zizhen Zhang** ®[3], **Yuta Kohro** ®[1,2], **Connor L. Anderson** ®[7], **Roger J. Thompson**[7], **Darren J. Derksen** ®[6], **Jose A. Morón** ®[4], **Gerald W. Zamponi** ®[3] **& Tuan Trang** ®[1,2] ✉

[1]Faculty of Veterinary Medicine, University of Calgary, Calgary, AB, Canada. [2]Department of Physiology and Pharmacology, Hotchkiss Brain Institute, University of Calgary, Calgary, AB, Canada. [3]Department of Clinical Neurosciences, Hotchkiss Brain Institute, University of Calgary, Calgary, AB, Canada.

[4]Department of Anesthesiology, Washington University School of Medicine, Washington University Pain Center, St. Louis, MO, USA. [5]Department of Anesthesiology & Perioperative Medicine, University of California Los Angeles, Los Angeles, CA, USA. [6]Department of Chemistry, University of Calgary, Calgary, AB, Canada. [7]Department of Cell Biology and Anatomy, Hotchkiss Brain Institute, University of Calgary, Calgary, AB, Canada. [8]These authors contributed equally: Charlie H. T. Kwok, Erika K. Harding, Nicole E. Burma. ✉e-mail: trangt@ucalgary.ca

