## [Peer Review File · Nature Communications]

Pannexin-1 channel inhibition alleviates opioid withdrawal by modulating locus coeruleus to spinal cord circuitryREVIEWER COMMENTS

Reviewer #1 (Remarks to the Author):

The manuscript by Kwok et al identifies a key role for microglial Panx1 in regulating the excitability of noradrenergic neurons in the locus coeruleus (LC) during opioid withdrawal. Their findings extend past work by the authors which identified an essential role for microglial Panx1 in mediating opioid withdrawal, but without identifying the specific circuits or target neurons responsible for the behavioral manifestations of opioid withdrawal. The LC, a key autonomic centre with dense opioid receptor expression, has previously been implicated in autonomic dysregulation tied to opioid withdrawal but how the LC becomes active in this context and the circuitry through which it affects physical and aversive sequelae of opioid withdrawal were not well defined.

The strength of the manuscript is the detailed approach used to establish the role of microglial Panx1 in modulating the activity of spinally projecting noradrenergic neurons in the LC. This includes microglial targeted conditional Panx1 deletion, pharmacological and chemogenetic approaches to silence spinal cord projecting LC neurons, and targeted ablation of NA neurons using DSP4 (especially convincing when delivered intrathecally).

Overall, the work is convincing and will appeal to the readership of the journal. Nevertheless, there are a number of issues that should be addressed:

1) While targeting microglia, Panx1 and NA neurons in the LC alleviates withdrawal symptoms, in most instances some withdrawal symptoms nevertheless remain (e.g. composite score of ~25 after LC lidocaine injection vs baseline of <10 in saline treated mice). Thus, LC may mediate some but not all withdrawal behaviors. For example, from suppl Fig 9 and 11 it's clear that piloerection and tremor/twitching do not respond to panx1 block by PRB/EB. Given this, in a supplement to Fig 2 it would be beneficial to provide scoring for separate symptoms. This would allow the authors to draw conclusions about which symptoms are reliant on spinally projecting LC neurons and which symptoms are not.

2) Fig 3C-E and suppl Fig 6: the results support that NLX increases AP firing in response to step depolarization in slices from MS treated mice and that such change requires microglia Panx1. Not reported is whether LC neurons have elevated spontaneous AP firing frequency (in the absence of current injection) or alternatively increased sEPSC freq, either of which would more directly support that LC excitability was increased in a circuit- or cell-intrinsic manner (related, was resting membrane potential or input resistance altered?). If not, this would support that the increased excitability of LC neurons (after NLX) only becomes manifest in response to excitatory drive from LC inputs (granted LC neurons may be more quiescent in slice). These possibilities should be discussed.

3) Fig 3F-G: ideally CNO/hM4Di effectiveness in silencing LC neurons should be confirmed by reduced cFos labelling (e.g. as shown for lidocaine in Fig 2B).

4) Fig 4G: a modest effect of DSP4 is seen when delivered systemically, but the effect on withdrawal scores is more robust when delivered intrathecally. The latter approach is more selective in targeting LC(spinal) neurons and therefore better suited. Given this, is the systemic DSP4 data even necessary? Authors could consider replacing systemic DSP4 data in Fig 4, with data shown in Suppl Fig 8 (effectiveness of intrathecal DSP4 in ablating projecting neurons is striking).

5) EG-2184: outcomes when using this probenecid derivative are mixed. While more potent in inhibiting Panx1 currents and dye flux, the inhibition is less as the concentration is increased (at least for dye flux; Fig 5D). Likewise, the suppression of withdrawal symptoms is more modest with EG (when compared to PRB) and the effectiveness is reduced at the highest concentration. In contrast,

EG is highly effective in reducing CPA and NE release after NLX. This should be acknowledged and discussed. Likewise, some brief additional information on the development of EG should be provided (e.g. on what basis was the carboxylic acid moiety targeted, was this strategy informed from recent cryo-EM Panx1 structures or was this part of a more comprehensive optimization effort). For suppl Fig 10, traces of Panx1 currents should be shown.

6) Last, I suggest adding a cartoon to illustrate the cell/molecular and circuit mechanisms identified by the authors. There are quite a few players involved and this would help summarize their major findings at-a-glance.

Minor:

1) P5, line 122: "the number of cFos-positive (cFos+) cells in several brain regions¹¹" additional supporting references should be provide as there is a large body of work on this topic (e.g. PMID: 35728954).

2) Supplementary Fig 1: only images of cFos in MS/VEH are shown. It would be more informative to show MS/VEH vs MS/TMX focusing on regions were differences between treatment groups were observed (mNACSh, CeA and LC).

3) Supplementary Fig 3: only 3-4 microglia are visible per image shown (although processes from microglia with cells bodies outside of the confocal plane are visible). N is stated in the fig legend for slices and mice, but how many microglia were counted per slice (or in total) should be indicated. Arrows could be added to highlight microglia positive for Panx1 vs those that are negative. As it stands is difficult to tell a difference between regions.

4) Fig 2: panel A is helpful, but would suggest adding the timing of drug injection to the schematic. This is stated in the methods, but it would be helpful to have this illustrated (e.g. mac1-sap for 3 days before NLX; lidocaine, 10panx, APR injected 1 hour before NLX).

5) Fig 2G: example image is not representative of the averaged data. In the example shown, mac-1-sap appears to have all but eliminated cFos staining.

Reviewer #2 (Remarks to the Author):

This is an extremely exciting manuscript that will be of interest to a wide scientific audience. This study represents a tour de force in methodological approaches to explore central Panx1 and its role in morphine withdrawal. The authors probe an interesting question and relatively novel circuit through the use of multiple challenging techniques. The work will be of high significance to the opioid withdrawal field; however, some additional evidence is required to best support the conclusions and claims of this work. First, expansion of the background surrounding Panx1, the LC, and the hypotheses made/tested will greatly enhance the cross-disciplinary relevance of this work. Most importantly, additional experiments and schematics are needed to consolidate the diverse experiments and ideas into one cohesive and convincing story. Currently the largest gap in this story is the behavioral relevance of the LC-> DH neurons to morphine withdrawal. The evidence for the role of the LC and Panx1 are well presented but the role of this specific neuronal population is not convincing with the single chemogenetic manipulation experiment of these neurons presented (the electrophysiology helps but is a touch underpowered due to the difficulty of these experiments). In addition to this gap, other areas for improvement include additional control experiments (Characterization/validation of Panx1 repopulation peripherally vs. centrally 28 days post-tamoxifen), minor data presentation edits (ex. outline the LC) and some alternative data analyses (ex. for Fos studies one individual data point is one

animal). Hopefully the following critiques may provide direction in how to improve and complete this fascinating story.

Major Comments:

-The authors need to provide evidence that Panx1 is conditionally knocked out at their experimental time frame. By 28 days post-tamoxifen both the central and peripheral microglia should have completely repopulated. Central microglial ablation studies have found repopulation within 10 days. Why was the morphine treatment not administered to these mice within days of tamoxifen treatment (28 days is extremely late)? Without conditional genetic data confirmation, it is completely unclear why morphine treatment did not begin more quickly after the tamoxifen-induced deletion of Panx1.

- To facilitate more interest/understanding from a wider scientific audience please elaborate further on the rationale for targeting Panx1 in the introduction and/or results. One sentence references the following seminal work but it can be further clarified. "Blocking microglial pannexin-1 channels alleviates morphine withdrawal in rodents."

-The Fos IHC (supplemental figure 2) and RNAscope + IHC (Supplemental Figure 3) are inappropriately quantified/powered. N=15 slices/2 mice is really N=2. It is appropriate to include at a minimum 4-5 mice/ group and each dot will indicate the average of multiple slices/ one animal. This is extremely important for the Fos analysis (supplemental figure 2) as the authors are making statistical inferences based on N=2. However, it appears the appropriate number of animals/ way to depict and quantify cFos was utilized for Figure 2. Thus, please maintain consistency and reevaluate supplemental Figures 2-3.

-What evidence do you provide for the specificity of EG-2184 at only PANX1?

-What is the relation of EG-2184 to LC-> DH neurons? These studies are entirely independent of one another. Can bath application of EG-2184 reduce LC->DH neuron excitability? Can it reduce Fos within these neurons after morphine withdrawal?

-a schematic is needed to elucidate the central idea of PANX1-expressing microglia exciting LC neurons to drive hyperexcitability and descending NE release. These ideas are independent throughout this manuscript and need to be unified at the end of the paper.

Additional Comments and Critiques:

-Supplemental figure 2 needs outlined brain regions (white) to better facilitate visualization.

-Figure 2 should have the entire LC outlined to appropriately display Fos counts/numbers

-Figure 3F's immunofluorescence image in the LC is not described in the figure legend and could also benefit from white tracing/outlining of the LC

-Supplemental Figure 3 is not convincing for Panx1/IBA1 co-localization. Most mRNA appears to be neuronal (IBA1-negative). Include arrows to indicate co-localization.

-Elaborate on 10panx and its validity as a selective Panx1 inhibitor.

- It would be valuable to include labeled videos with examples of tremors, jumping, and wet-dog shakes.

-Perhaps the most exciting piece of data is the manipulation/modulation of the descending spinal input. Can you expand your hypothesis/introduction to this experiment and your rationale for first looking at descending LC -> DH inputs?

-In relation to the Figure 3 retrobead study- can you provide a mean and SEM of the number of LC ->DH neurons/animal. This is some of the first anatomical descriptions of this circuit and it would be valuable beyond percentages for anatomists and other scientists to understand the density of this circuit.

-Figure 3's organization of D-E and F-G should be reorganized to facilitate ease of reading (always left to right).

-For the electrophysiological characterization of LC->DH neurons in Figure 3 - can you report (can be supplemental) additional parameters from your recordings such as 1st spike latency, rheobase, and amplitude?

- Does silencing LCspinal neurons affect motor locomotion/coordination that may interfere with the withdrawal behaviors?

-Can bath or puff application of 10panx reduce naloxone-induced increases in neuronal firing in LCspinal neurons?

REVIEWER COMMENTS

Reviewer #1 (Remarks to the Author):

The manuscript by Kwok et al identifies a key role for microglial Panx1 in regulating the excitability of noradrenergic neurons in the locus coeruleus (LC) during opioid withdrawal. Their findings extend past work by the authors which identified an essential role for microglial Panx1 in mediating opioid withdrawal, but without identifying the specific circuits or target neurons responsible for the behavioral manifestations of opioid withdrawal. The LC, a key autonomic centre with dense opioid receptor expression, has previously been implicated in autonomic dysregulation tied to opioid withdrawal but how the LC becomes active in this context and the circuitry through which it affects physical and aversive sequelae of opioid withdrawal were not well defined.

The strength of the manuscript is the detailed approach used to establish the role of microglial Panx1 in modulating the activity of spinally projecting noradrenergic neurons in the LC. This includes microglial targeted conditional Panx1 deletion, pharmacological and chemogenetic approaches to silence spinal cord projecting LC neurons, and targeted ablation of NA neurons using DSP4 (especially convincing when delivered intrathecally).

Overall, the work is convincing and will appeal to the readership of the journal. Nevertheless, there are a number of issues that should be addressed:

1) While targeting microglia, Panx1 and NA neurons in the LC alleviates withdrawal symptoms, in most instances some withdrawal symptoms nevertheless remain (e.g. composite score of ~25 after LC lidocaine injection vs baseline of <10 in saline treated mice). Thus, LC may mediate some but not all withdrawal behaviors. For example, from suppl Fig 9 and 11 it's clear that piloerection and tremor/twitching do not respond to panx1 block by PRB/EB. Given this, in a supplement to Fig 2 it would be beneficial to provide scoring for separate symptoms. This would allow the authors to draw conclusions about which symptoms are reliant on spinally projecting LC neurons and which symptoms are not.

2) Fig 3C-E and suppl Fig 6: the results support that NLX increases AP firing in response to step depolarization in slices from MS treated mice and that such change requires microglia Panx1. Not reported is whether LC neurons have elevated spontaneous AP firing frequency (in the absence of current injection) or alternatively increased sEPSC freq, either of which would more directly support that LC excitability was increased in a circuit- or cell-intrinsic manner (related, was resting membrane potential or input resistance altered?). If not, this would support that the increased excitability of LC neurons (after NLX) only becomes manifest in response to excitatory drive from LC inputs (granted LC neurons may be more quiescent in slice). These possibilities should be discussed.

3) Fig 3F-G: ideally CNO/hM4Di effectiveness in silencing LC neurons should be confirmed by reduced cFos labelling (e.g. as shown for lidocaine in Fig 2B).

4) Fig 4G: a modest effect of DSP4 is seen when delivered systemically, but the effect on withdrawal scores is more robust when delivered intrathecally. The latter approach is more selective in targeting LC(spinal) neurons and therefore better suited. Given this, is the systemic DSP4 data even necessary? Authors could consider replacing systemic DSP4 data in Fig 4, with data shown in Suppl Fig 8 (effectiveness of intrathecal DSP4 in ablating projecting neurons is striking).

5) EG-2184: outcomes when using this probenecid derivative are mixed. While more potent in

inhibiting Panx1 currents and dye flux, the inhibition is less as the concentration is increased (at least for dye flux; Fig 5D). Likewise, the suppression of withdrawal symptoms is more modest with EG (when compared to PRB) and the effectiveness is reduced at the highest concentration. In contrast, EG is highly effective in reducing CPA and NE release after NLX. This should be acknowledged and discussed. Likewise, some brief additional information on the development of EG should be provided (e.g. on what basis was the carboxylic acid moiety targeted, was this strategy informed from recent cryo-EM Panx1 structures or was this part of a more comprehensive optimization effort). For suppl Fig 10, traces of Panx1 currents should be shown.

6) Last, I suggest adding a cartoon to illustrate the cell/molecular and circuit mechanisms identified by the authors. There are quite a few players involved and this would help summarize their major findings at-a-glance.

Minor:

1) P5, line 122: “the number of cFos-positive (cFos+) cells in several brain regions11” additional supporting references should be provide as there is a large body of work on this topic (e.g. PMID: 35728954).

2) Supplementary Fig 1: only images of cFos in MS/VEH are shown. It would be more informative to show MS/VEH vs MS/TMX focusing on regions were differences between treatment groups were observed (mNAcSh, CeA and LC).

3) Supplementary Fig 3: only 3-4 microglia are visible per image shown (although processes from microglia with cells bodies outside of the confocal plane are visible). N is stated in the fig legend for slices and mice, but how many microglia were counted per slice (or in total) should be indicated. Arrows could be added to highlight microglia positive for Panx1 vs those that are negative. As it stands is difficult to tell a difference between regions.

4) Fig 2: panel A is helpful, but would suggest adding the timing of drug injection to the schematic. This is stated in the methods, but it would be helpful to have this illustrated (e.g. mac1-sap for 3 days before NLX; lidocaine, 10panx, APR injected 1 hour before NLX).

5) Fig 2G: example image is not representative of the averaged data. In the example shown, mac-1-sap appears to have all but eliminated cFos staining.

Reviewer #2 (Remarks to the Author):

This is an extremely exciting manuscript that will be of interest to a wide scientific audience. This study represents a tour de force in methodological approaches to explore central Panx1 and its role in morphine withdrawal. The authors probe an interesting question and relatively novel circuit through the use of multiple challenging techniques. The work will be of high significance to the opioid withdrawal field; however, some additional evidence is required to best support the conclusions and claims of this work. First, expansion of the background surrounding Panx1, the LC, and the hypotheses made/tested will greatly enhance the cross-disciplinary relevance of this work. Most importantly, additional experiments and schematics are needed to consolidate the diverse experiments and ideas into one cohesive and convincing story. Currently the largest gap in this story is the behavioral relevance of the LC-> DH neurons to morphine withdrawal. The evidence for the role of the LC and Panx1 are well presented but the role of this specific neuronal population is not convincing with the single

chemogenetic manipulation experiment of these neurons presented (the electrophysiology helps but is a touch underpowered due to the difficulty of these experiments). In addition to this gap, other areas for improvement include additional control experiments (Characterization/validation of Panx1 repopulation peripherally vs. centrally 28 days post-tamoxifen), minor data presentation edits (ex. outline the LC) and some alternative data analyses (ex. for Fos studies one individual data point is one animal). Hopefully the following critiques may provide direction in how to improve and complete this fascinating story.

Major Comments:

-The authors need to provide evidence that Panx1 is conditionally knocked out at their experimental time frame. By 28 days post-tamoxifen both the central and peripheral microglia should have completely repopulated. Central microglial ablation studies have found repopulation within 10 days. Why was the morphine treatment not administered to these mice within days of tamoxifen treatment (28 days is extremely late)? Without conditional genetic data confirmation, it is completely unclear why morphine treatment did not begin more quickly after the tamoxifen-induced deletion of Panx1.

- To facilitate more interest/understanding from a wider scientific audience please elaborate further on the rationale for targeting Panx1 in the introduction and/or results. One sentence references the following seminal work but it can be further clarified. "Blocking microglial pannexin-1 channels alleviates morphine withdrawal in rodents."

-The Fos IHC (supplemental figure 2) and RNAscope + IHC (Supplemental Figure 3) are inappropriately quantified/powerd. N=15 slices/2 mice is really N=2. It is appropriate to include at a minimum 4-5 mice/ group and each dot will indicate the average of multiple slices/ one animal. This is extremely important for the Fos analysis (supplemental figure 2) as the authors are making statistical inferences based on N=2. However, it appears the appropriate number of animals/ way to depict and quantify cFos was utilized for Figure 2. Thus, please maintain consistency and reevaluate supplemental Figures 2-3.

-What evidence do you provide for the specificity of EG-2184 at only PANX1?

-What is the relation of EG-2184 to LC-> DH neurons? These studies are entirely independent of one another. Can bath application of EG-2184 reduce LC->DH neuron excitability? Can it reduce Fos within these neurons after morphine withdrawal?

-a schematic is needed to elucidate the central idea of PANX1-expressing microglia exciting LC neurons to drive hyperexcitability and descending NE release. These ideas are independent throughout this manuscript and need to be unified at the end of the paper.

Additional Comments and Critiques:

-Supplemental figure 2 needs outlined brain regions (white) to better facilitate visualization.

-Figure 2 should have the entire LC outlined to appropriately display Fos counts/numbers

-Figure 3F's immunofluorescence image in the LC is not described in the figure legend and could also benefit from white tracing/outlining of the LC

-Supplemental Figure 3 is not convincing for Panx1/IBA1 co-localization. Most mRNA appears to be neuronal (IBA1-negative). Include arrows to indicate co-localization.

-Elaborate on 10panx and its validity as a selective Panx1 inhibitor.

- It would be valuable to include labeled videos with examples of tremors, jumping, and wet-dog shakes.

-Perhaps the most exciting piece of data is the manipulation/modulation of the descending spinal input. Can you expand your hypothesis/introduction to this experiment and your rationale for first looking at descending LC -> DH inputs?

-In relation to the Figure 3 retrobead study- can you provide a mean and SEM of the number of LC ->DH neurons/animal. This is some of the first anatomical descriptions of this circuit and it would be valuable beyond percentages for anatomists and other scientists to understand the density of this circuit.

-Figure 3's organization of D-E and F-G should be reorganized to facilitate ease of reading (always left to right).

-For the electrophysiological characterization of LC->DH neurons in Figure 3 - can you report (can be supplemental) additional parameters from your recordings such as 1st spike latency, rheobase, and amplitude?

- Does silencing LCspinal neurons affect motor locomotion/coordination that may interfere with the withdrawal behaviors?

-Can bath or puff application of 10panx reduce naloxone-induced increases in neuronal firing in LCspinal neurons?

Response to Reviewer Comments

We thank the reviewers for providing insightful and constructive comments that increased the scope and clarity of our manuscript. We are pleased that the reviewers were enthusiastic about the novelty of our study and its impact for understanding the underlying causes, and the potential treatment, of opioid withdrawal. In response to the comments, we have performed additional experiments and revised the manuscript to fully address the points raised. The new data strengthen our original conclusion that spinally projecting locus coeruleus (LC) noradrenergic output is critical for opioid withdrawal. Notable new experimental findings and revisions to the manuscript include:

- ***RNAscope***: We have increased the sample size to n=4 per group for RNAscope fluorescent in situ hybridization experiments. The additional data support our original observation that within the LC, Panx1 transcripts are expressed in microglia. (**Supplementary Fig. 3**)
- ***cFos staining***: There is additional quantification of cFos across key brain regions in the revised manuscript. The data confirm an increase in cFos+ neurons within the LC after naloxone-induced opioid withdrawal and that the response is blunted in microglia Panx1-deficient mice. (**Supplementary Fig. 2**)
- ***iDREADD suppresses cFos response in the LC***: We have new data showing that selective DREADD inhibition of spinally projecting LC neurons suppresses opioid withdrawal induced cFos response. These experiments complement the existing iDREADD data indicating that silencing LC^{spinal} neurons alleviates both physical and aversive opioid withdrawal behaviours. (**Supplementary Fig. 11**)
- ***LC hyperexcitability is suppressed by EG-2184***: In whole cell recordings of LC brain slices, we show that bath application of EG-2184 suppresses hyperexcitability of LC^{spinal} neurons during opioid withdrawal. These experiments provide direct evidence that EG-2184 modulates LC output during naloxone induced opioid withdrawal, which strengthens the utility of this new compound in reducing opioid withdrawal and cue-induced reinstatement behaviours. (**Fig. 5H-J**)

Re: Comments from Reviewer 1

We thank the reviewer for commenting that our work is “convincing and will appeal to the readership of the journal.” We also appreciate the reviewer’s suggestions for additional experiments that have increased the scope of our study.

1) While targeting microglia, Panx1 and NA neurons in the LC alleviates withdrawal symptoms, in most instances some withdrawal symptoms nevertheless remain (e.g. composite score of ~25 after LC lidocaine injection vs baseline of <10 in saline treated mice). Thus, LC may mediate some but not all withdrawal behaviors. For example, from suppl Fig 9 and 11 it’s clear that piloerection and tremor/twitching do not respond to panx1 block by PRB/EB. Given this, in a supplement to Fig 2 it would be beneficial to provide scoring for separate symptoms.

In the revised manuscript, we include individual withdrawal behaviours in **Supplementary Figures 5, 7, 8, and 9** for each intervention in which a cumulative withdrawal score is reported.

2) Fig 3C-E and suppl Fig 6: the results support that NLX increases AP firing in response to step depolarization in slices from MS treated mice and that such change requires microglia Panx1. Not reported is whether LC neurons have elevated spontaneous AP firing frequency (in the absence of

current injection) or alternatively increased sEPSC freq, either of which would more directly support that LC excitability was increased in a circuit- or cell-intrinsic manner (related, was resting membrane potential or input resistance altered?). If not, this would support that the increased excitability of LC neurons (after NLX) only becomes manifest in response to excitatory drive from LC inputs (granted LC neurons may be more quiescent in slice). These possibilities should be discussed.

Thank you for this suggestion, we have revised the Discussion to address the possible causes of the increased LC excitability during opioid withdrawal (**Page 13, lines 299 – 310**):

“Our study answers an important longstanding question – how and whether the LC is dysregulated during opioid withdrawal^{9,10,11,30,31}. In a unique population of spinally projecting neurons within the LC, we show that intrinsic excitability is not impacted by repeated morphine treatment. However, application of naloxone unmasks a hyperexcitability in these LC^{spinal} neurons, that when pharmacologically or chemogenetically silenced, reduces opioid withdrawal and CPA behaviours. The LC^{spinal} recordings were conducted in preparations that were not synaptically isolated (i.e., there was no bath application of glutamatergic and GABAergic inhibitors). The hyperexcitability could be caused by alterations in the intrinsic firing properties of the recorded spinally projecting LC neuron, due to increased excitatory and/or decreased inhibitory synaptic input. Indeed, the LC receives inputs from and projects to many brain regions including the ventral tegmental area, nucleus accumbens, periaqueductal gray, BNST, and thalamus, which influence opioid analgesia, withdrawal, and drug seeking behaviours.”

We note that although spontaneous EPSP frequency was recorded, these recordings were not sufficiently long to quantitatively ascertain alterations in excitatory synaptic inputs. Therefore, we cannot discern among the aforementioned possibilities; future experiments focusing on synaptic versus non synaptic mechanisms will be needed to address this issue in detail.

3) Fig 3F-G: ideally CNO/hM4Di effectiveness in silencing LC neurons should be confirmed by reduced cFos labelling (e.g. as shown for lidocaine in Fig 2B).

We performed additional experiments in which AAV_{retro}-P_{gk}-cre was injected spinally to express cre in spinal cord projecting neurons. Cre-dependent hM4Di was then injected into the LC so that only LC^{spinal} neurons express hM4Di. Our new data shows that cFos response during opioid withdrawal is blunted by selective DREADD inhibition of LC^{spinal} neurons (**Supplementary Fig. 11A,B**). These results strengthen our existing iDREADD data, indicating that silencing LC^{spinal} neurons alleviates both physical and aversive opioid withdrawal behaviours. Importantly, the findings strengthen our conclusion that the LC is a critical anatomical hub for opioid withdrawal.

4) Fig 4G: a modest effect of DSP4 is seen when delivered systemically, but the effect on withdrawal scores is more robust when delivered intrathecally. The latter approach is more selective in targeting LC(spinal) neurons and therefore better suited. Given this, is the systemic DSP4 data even necessary? Authors could consider replacing systemic DSP4 data in Fig 4, with data shown in Suppl Fig 8 (effectiveness of intrathecal DSP4 in ablating projecting neurons is striking).

We agree and have reorganized **Figure 4**. The systemic DSP4 data is now presented in Supplementary **Figure 13** and the intrathecal DSP data in **Figure 4**.

5) EG-2184: outcomes when using this probenecid derivative are mixed. While more potent in

inhibiting Panx1 currents and dye flux, the inhibition is less as the concentration is increased (at least for dye flux; Fig 5D). Likewise, the suppression of withdrawal symptoms is more modest with EG (when compared to PRB) and the effectiveness is reduced at the highest concentration. In contrast, EG is highly effective in reducing CPA and NE release after NLX. This should be acknowledged and discussed. Likewise, some brief additional information on the development of EG should be provided (e.g. on what basis was the carboxylic acid moiety targeted, was this strategy informed from recent cryo-EM Panx1 structures or was this part of a more comprehensive optimization effort). For suppl Fig 10, traces of Panx1 currents should be shown.

In the revised manuscript we provide more rationale and information on the development of EG-2184: For improved potency and delivery, we synthesized the compound EG-2184 by replacing the carboxylic acid of probenecid with a tertiary hexafluoryl alcohol, which increases lipophilicity while maintaining the hydrogen bond donor ability of probenecid (**Page 10, Lines 241-243**). The synthesis work was completed before the cryo-EM Panx1 structure was published. Therefore, the design of this compound was based on a targeted chemical structure optimization strategy.

In the revised discussion, we also state that: Although EG-2184 is more potent than probenecid at inhibiting Panx1 currents and dye flux, the effective dose of this compound and its delivery in oral or other clinical formulations will require further pharmacokinetic and pharmacodynamic optimization (**Page 14, Lines 336-338**).

Panx1 current traces in the presence of probenecid and EG-2184 are presented in **Supplementary Fig. 15**.

6) Last, I suggest adding a cartoon to illustrate the cell/molecular and circuit mechanisms identified by the authors. There are quite a few players involved and this would help summarize their major findings at-a-glance.

We have included a summary figure to illustrate the LC-spinal circuitry and its key players (**Supplementary Fig. 21**).

Minor Comments:

1) P5, line 122: “the number of cFos-positive (cFos+) cells in several brain regions” additional supporting references should be provide as there is a large body of work on this topic (e.g. PMID: 35728954).

Additional references have been (**Page 5, Line 106**).

2) Supplementary Fig 1: only images of cFos in MS/VEH are shown. It would be more informative to show MS/VEH vs MS/TMX focusing on regions where differences between treatment groups were observed (mNAcSh, CeA and LC).

Supplementary Figure 2 has been revised to include representative cFos images for CTR/VEH, MS/VEH, and MS/TMX within the LC. We have increased sample size from n=2 per brain region to n=5 (CTR/VEH), n=7 (MS/VEH), and n=6 (MS/TMX). In increasing sample size, the mNAcSh and LC retained a significant microglia Panx1-dependent reduction in withdrawal induced cFos expression. This change is reflected in the revised manuscript.

3) Supplementary Fig 3: only 3-4 microglia are visible per image shown (although processes from microglia with cells bodies outside of the confocal plane are visible). N is stated in the fig legend for slices and mice, but how many microglia were counted per slice (or in total) should be indicated. Arrows could be added to highlight microglia positive for Panx1 vs those that are negative. As it stands is difficult to tell a difference between regions.

We have increased the sample size to n=4 mice per group in each brain region. The number of microglia counted is now stated in the figure legends and arrows have been added to highlight microglia positive for Panx1.

4) Fig 2: panel A is helpful, but would suggest adding the timing of drug injection to the schematic. This is stated in the methods, but it would be helpful to have this illustrated (e.g. mac1-sap for 3 days before NLX; lidocaine, 10panx, APR injected 1 hour before NLX).

Schematics have been added to clarify timeline for injections in Fig. 2A.

5) Fig 2G: example image is not representative of the averaged data. In the example shown, mac-1-sap appears to have all but eliminated cFos staining.

A more representative image has been added.

Re: Comments from Reviewer 2

Thank you for taking time to review our manuscript. We appreciate the comments that our manuscript will be of “interest to a wide scientific audience” and that it is “a tour de force” in methodological approaches. The comments were insightful, constructive, and have improved the clarity and breadth of the manuscript.

Major Comments:

1) The authors need to provide evidence that Panx1 is conditionally knocked out at their experimental time frame. By 28 days post-tamoxifen both the central and peripheral microglia should have completely repopulated. Central microglial ablation studies have found repopulation within 10 days. Why was the morphine treatment not administered to these mice within days of tamoxifen treatment (28 days is extremely late)? Without conditional genetic data confirmation, it is completely unclear why morphine treatment did not begin more quickly after the tamoxifen-induced deletion of Panx1.

The reviewer is correct that microglia repopulate within 10 days after ablation and have included a Discussion to improve clarity (**Page14, lines 317-327**). For Mac1-saporin experiments, mice were treated with morphine for 5 days. On days 2-4 of morphine treatment, Mac1-saporin was locally injected into the LC, resulting in microglia depletion. Thus, microglia were depleted at the time of naloxone challenge, resulting in attenuated morphine withdrawal behaviours. By contrast, in Cx3cr1-Cre^{ERT}::Panx1^{flx/flx} mice tamoxifen was administered to induce Cre-recombination in CX₃CR₁ cells, causing specific conditional knockdown of Panx1. *Unlike Mac1-saporin, these were not microglia ablation experiments.* We therefore waited 28 days to allow for the normal process of repopulation of peripheral CX₃CR₁ cells; central CX₃CR₁ cells (e.g. microglia) will remain Panx1-deficient (Parkhurst

et al., 2013). Indeed, we previously detailed in these mice that: 1) Cre is expressed in microglia, and 2) both Panx1 expression and function are significantly reduced in microglia after tamoxifen treatment (Burma et al., 2017, Nature Medicine). In a separate unpublished study, we have further confirmed that Panx1 function in spinal microglia remains impaired 28 days after tamoxifen treatment.

2) To facilitate more interest/understanding from a wider scientific audience please elaborate further on the rationale for targeting Panx1 in the introduction and/or results. One sentence references the following seminal work but it can be further clarified. “Blocking microglial pannexin-1 channels alleviates morphine withdrawal in rodents.”

Thank you for this suggestion. We revised the introduction to provide better contextual our current and previous study: “Since microglial pannexin-1 (Panx1) channel activation is critical for long-term synaptic facilitation in the spinal cord during opioid withdrawal, we asked whether this mechanism also has a supraspinal site of action. In examining the LC, we uncovered a specific top-down LC to spinal circuit that is crucial for the physical and aversive sequelae of opioid withdrawal. We show that the aberrant output of spinally projecting LC neurons during opioid withdrawal critically requires microglial Panx1 activation, providing a unifying opioid withdrawal mechanism in distinct spinal and supraspinal centres.” (**Page 3, Lines 75-81**).

3) The Fos IHC (supplemental figure 2) and RNAscope + IHC (Supplemental Figure 3) are inappropriately quantified/powered. N=15 slices/2 mice is really N=2. It is appropriate to include at a minimum 4-5 mice/ group and each dot will indicate the average of multiple slices/ one animal. This is extremely important for the Fos analysis (supplemental figure 2) as the authors are making statistical inferences based on N=2. However, it appears the appropriate number of animals/ way to depict and quantify cFos was utilized for Figure 2. Thus, please maintain consistency and reevaluate supplemental Figures 2-3.

For cFos of the various brain regions, sample sizes are increased from n=2 to n=5 (CTR/VEH), n=7 (MS/VEH), and n=6 (MS/TMX) (**Supplementary Fig. 2**). RNAscope + IHC sample size has also been increased to n=4 per group (**Supplementary Fig. 3**).

4) What evidence do you provide for the specificity of EG-2184 at only PANX1?

This is an excellent question. Without a comprehensive screen, which is cost and time prohibitive, we have refrained from making claims or statements about the specificity of EG-2184 for Panx1. Instead, our dye flux and patch clamp data show that EG-2184 is significantly more potent than probenecid at inhibiting Panx1. We have additional lines of evidence indicating that EG-2184 has a Panx1 mechanism of action (**Supplementary Fig. 18**):

1. In microglial Panx1-deficient mice, which display attenuated withdrawal behaviours, administration of EG-2184 did not produce a further reduction in morphine withdrawal (**Supplementary Fig. 18A,B**). This suggests that EG-2184 mediated suppression of opioid withdrawal behaviours is dependent upon its actions on microglia Panx1.
2. P2X7 receptor activation is a core mechanism for opening Panx1 channels. We have new data that P2X7R-mediated calcium responses in BV2 microglia-like cells are *not* affected by EG-2184 (10 nM) at a concentration that significantly blocked Panx1-mediated currents and dye flux (**Supplementary Fig. 18C**). Our new data demonstrate that EG-2184 blocks Panx1 activity without off-target inhibition of P2X7 receptors.

5) What is the relation of EG-2184 to LC-> DH neurons? These studies are entirely independent of one another. Can bath application of EG-2184 reduce LC->DH neuron excitability? Can it reduce Fos within these neurons after morphine withdrawal?

Thank you for this suggestion. In the revised manuscript, we include LC brain slice recordings from morphine dependent mice showing that bath application of EG-2184 abolishes LC^{spinal} neuron hyperexcitability following naloxone challenge (**Fig 5H-J**). The new data are consistent with our finding that EG-2184 blunts LC cFos response during opioid withdrawal. Thus, our collective findings indicate that inhibition of LC^{spinal} excitability is a key mechanism of EG-2184 action.

6) a schematic is needed to elucidate the central idea of PANX1-expressing microglia exciting LC neurons to drive hyperexcitability and descending NE release. These ideas are independent throughout this manuscript and need to be unified at the end of the paper.

We have included a schematic to better illustrate the LC-spinal circuitry and its key cellular players in opioid withdrawal (**Supplementary Fig. 21**).

Additional Comments and Critiques:

1) Supplemental figure 2 needs outlined brain regions (white) to better facilitate visualization.

Brain regions are now more clearly outlined (**Supplementary Fig. 2**). We have simplified the figure and included the three representative groups for LC, as suggested by Reviewer 1.

2) Figure 2 should have the entire LC outlined to appropriately display Fos counts/numbers

Figure 2 has been revised to outline the LC more clearly.

3) Figure 3F's immunofluorescence image in the LC is not described in the figure legend and could also benefit from white tracing/outlining of the LC.

A legend has been added and the LC outlined in the revised **Figure 3F**.

4) Supplemental Figure 3 is not convincing for Panx1/IBA1 co-localization. Most mRNA appears to be neuronal (IBA1-negative). Include arrows to indicate co-localization.

Image quality has been improved and arrows added to show co-localization in **Supplemental Fig. 3**.

5) Elaborate on 10panx and its validity as a selective Panx1 inhibitor.

¹⁰panx is a 10-amino acid peptide with sequence WRQAAFVDSY corresponding to residues 74-83 of the Panx1 extracellular loop 1, of which W74 and R75 are essential for gating of the Panx1 channel (Pelegriin & Surprenant, 2006; Michalski et al., 2020; Rusiecka et al., 2022). ¹⁰panx is highly selective for pannexin-1 and does not inhibit P2X receptors (Pelegriin and Surprenant, 2009; Burma et al., 2017). A comprehensive study by Wang et al. (2007) evaluated the actions of ¹⁰panx on Panx1 and multiple connexin family proteins. They found a modest inhibition of connexin-46 (Cx46) by ¹⁰panx at 200 μM.

However, Cx46 is expressed only in the lens of the eye and forms gap junctions, not hemichannels. Furthermore, ¹⁰panx inhibition of Cx46 occurred under divalent cation free solutions. In our current and previous study, we provide converging pharmacological, genetic, and cellular data implicating Panx1 in opioid withdrawal and consistent with ¹⁰panx inhibition of Panx1. We also used a scrambled ¹⁰panx control peptide, which did not affect Panx1 activity and did not impact opioid withdrawal behaviours.

We have included some background in the revised Results to provide more clarity on the use of ¹⁰panx (**Page 6, lines 144-147**).

6) It would be valuable to include labeled videos with examples of tremors, jumping, and wet-dog shakes.

After consulting with our Animal Care Committee Chair and Head Veterinarian, we were advised to not upload videos of animals undergoing opioid withdrawal because of potential concerns about misrepresentation of the videos if they were shared or distributed to the public. We have videos of the withdrawal behaviours and can make them readily available to the reviewers, but respectfully request that they not be available for the public. We thank the reviewer for their understanding.

7) Perhaps the most exciting piece of data is the manipulation/modulation of the descending spinal input. Can you expand your hypothesis/introduction to this experiment and your rationale for first looking at descending LC -> DH inputs?

Thank you for this suggestion and that of comment #2. The introduction has been revised to contextualize this study and to provide a clearer rationale for focusing on the LC to spinal circuitry in opioid withdrawal.

8) In relation to the Figure 3 retrobead study- can you provide a mean and SEM of the number of LC ->DH neurons/animal. This is some of the first anatomical descriptions of this circuit and it would be valuable beyond percentages for anatomists and other scientists to understand the density of this circuit.

This is a good point. We now provide a quantification in **Supplementary Fig. 12** and a description in the Results (**Pages 8-9, lines 197-210**).

9) Figure 3's organization of D-E and F-G should be reorganized to facilitate ease of reading (always left to right).

Figure 3 has been reorganized to improve readability.

10) For the electrophysiological characterization of LC->DH neurons in Figure 3 - can you report (can be supplemental) additional parameters from your recordings such as 1st spike latency, rheobase, and amplitude?

In the revised manuscript, we have included additional electrophysiological characterization of LC-spinal neurons (**Supplementary Fig. 10**). We have also provided a response to Reviewer 1 (question 2).

11) Does silencing LCspinal neurons affect motor locomotion/coordination that may interfere with the withdrawal behaviors?

We found that iDREADD selectively silencing of LC^{spinal} neurons did not significantly affect overall locomotion (i.e. distance travelled and velocity) in CNO vs control mice (**Supplementary Fig. 11B,C**). Furthermore, locomotion and thermal tail-flick responses are not altered by treatment with EG-2184 and remained intact in microglia Panx1-deficient mice, indicating locomotion and spinal reflex/thermal sensory responses are not impaired when blocking Panx1 channels (Burma et al., 2017a, b).

12) Can bath or puff application of 10panx reduce naloxone-induced increases in neuronal firing in LC^{spinal} neurons?

Thank you for this comment. Reviewer 1 had a similar comment about bath application of a Panx1 blocking compound and whether there is a direct consequence on naloxone induced LC^{spinal} excitability. For these experiments, we tested EG-2184 because of its high potency in blocking Panx1 activity. Our new data shows that bath application of EG-2184 suppresses LC^{spinal} neuron hyperexcitability following naloxone challenge in morphine dependent mice (**Fig. 5H-J**). This provides direct evidence for the actions of EG-2184 in the LC and it is consistent with EG-2184 suppression of LC cFos response.

REVIEWERS' COMMENTS

Reviewer #1 (Remarks to the Author):

The study by Kwok et al. has been further strengthened by the extensive revisions undertaken. This includes important additional experiments, more rigorous analysis and detailed presentation of their extensive data sets. Study and experiment rationale has been improved and major findings are now more critically discussed. The rebuttal letter comprehensively addresses all of my major concerns. Overall, the manuscript is very convincing and represents an important advancement to our understanding of opioid withdrawal and its potential treatment.